# Twofold rigidity activates ultralong organic high-temperature phosphorescence

Kaijun Chen[1,4], Yongfeng Zhang[2,4], Yunxiang Lei [1] ✉, Wenbo Dai[1], Miaochang Liu[1], Zhengxu Cai [2], Huayue Wu[1], Xiaobo Huang [1] ✉ & Xiang Ma [3] ✉

A strategy is pioneered for achieving high-temperature phosphorescence using planar rigid molecules as guests and rigid polymers as host matrix. The planar rigid configuration can resist the thermal vibration of the guest at high temperatures, and the rigidity of the matrix further enhances the high-temperature resistance of the guest. The doped materials exhibit an afterglow of 40 s at 293 K, 20 s at 373 K, 6 s at 413 K, and a 1 s afterglow at 433 K. The experimental results indicate that as the rotational ability of the groups connected to the guests gradually increases, the high-temperature phosphorescence performance of the doped materials gradually decreases. In addition, utilizing the property of doped materials that can emit phosphorescence at high temperatures and in high smoke, the attempt is made to use organic phosphorescence materials to identify rescue workers and trapped personnel in fires.

Organic phosphorescence materials with ultralong afterglow are an active area of research because of their advantages in information encryption, organic light-emitting diodes, anti-counterfeiting, luminescent sensing, and bioimaging[1–15]. Organic compounds, which lack metal atoms and exhibit substantial molecular motion, were once considered unlikely to emit room-temperature phosphorescence (RTP). Various strategies, such as introducing an aromatic carbonyl or halogen atoms, forming clusters, H-aggregation, crystal engineering, and doping small organic molecules into a rigid matrix, have been proposed to develop organic compounds capable of emitting RTP[16–25]. Therefore, in recent years, the phosphorescence performance of organic materials has substantially improved.

After organic materials absorb excitation energy, the electrons transition from the ground state to the excited state. However, the excited state energy is unstable and readily returns to the ground state through the non-radiative process due to the molecular motion[26–28]. High temperature increases thermal molecular motion. Therefore, excitons (especially triplet excitons formed by intersystem crossing of singlet excitons) are sensitive to temperature and prone to thermal deactivation, resulting in the rapid termination of the afterglow at high temperature[29–31]. The fact that phosphorescence emission is not resistant to high temperatures substantially hinders the application range of corresponding organic materials. Therefore, developing a strategy to construct ultralong organic high-temperature phosphorescence (HTP) materials that can maintain phosphorescence even under high thermal conditions is critical.

Using covalent copolymers, Xie et al. found that multiple hydrogen bonds in copolymers can substantially inhibit non-radiative transitions, resulting in materials with an afterglow at 413 K[32,33]. Chen et al. self-assembled a two-dimensional organic material using hydrogen bonds, and detected a phosphorescence signal at 473 K[34]. Xu et al. obtained a non-conjugated polymer through free-radical polymerization with an afterglow of 3 s at 353 K[35]. In addition, the ionic bonding cross-linking polymers prepared by An et al. and the quaternary phosphonium derivative-based polymers designed by Zhao et al. all exhibited phosphorescence emission at high temperatures[36,37].

[1]School of Chemistry and Materials Engineering, Wenzhou University, Wenzhou 325035, PR China. [2]School of Materials Science & Engineering, Beijing Institute of Technology, 10081 Beijing, PR China. [3]Key Laboratory for Advanced Materials and Feringa Nobel Prize Scientist Joint Research Center, Frontiers Science Center for Materiobiology and Dynamic Chemistry, School of Chemistry and Molecular Engineering, East China University of Science and Technology, Meilong Road 130, Shanghai 200237, PR China. [4]These authors contributed equally: Kaijun Chen, Yongfeng Zhang. ✉e-mail: yunxianglei@wzu.edu.cn; xiaobhuang@wzu.edu.cn; maxiang@ecust.edu.cn

Moreover, Li et al. constructed a series of HTP materials with excellent luminescent properties, and studied intermolecular and intramolecular interactions at various temperatures, as well as the resulting changes in phosphorescence wavelength[38]. The aforementioned findings provide important references for designing HTP materials. However, it remains a challenge to propose facile and feasible molecular design strategy for preparing HTP materials to further extend the application potential.

Intramolecular motion, especially thermal motion at high temperatures, results in the inactivation of excitons and consequent termination of afterglow. Therefore, inhibiting molecular motion is a crucial approach to endow organic materials with HTP properties[29,30]. The host–guest doped method refers to doping a small amount of guest molecules into the host matrix to impart the mixture with RTP activity. The host provides a rigid environment that suppresses nonradiative transitions of excitons[39–42]. However, the phosphorescence performance of doped materials also significantly depends on the guest molecules. Therefore, enhancing the resistance of guest molecules to high-temperature vibrations can also contribute to achieving HTP activity in organic materials (Fig. 1).

Accordingly, a rigid molecule 9H-dibenzo[a,c]carbazole (BCZ) with a nearly completely planar molecular configuration was selected as the guest (Fig. 2a). The planar configuration can minimize vibration and rotation of molecules. In addition, the rigid configuration is conducive to increasing the phosphorescence lifetime and prolonging the afterglow time[43]. Polyvinyl pyrrolidone (PVP) with a rigid structure and high glass transition temperature ($T_g$) was selected as the host to further inhibit the molecular motion of the guest. This twofold rigidity resulted in a doped material BCZ/PVP that exhibited an ultralong afterglow of 40 s at 293 K, a strong afterglow of up to 20 s at 373 K, a bright afterglow of about 6 s at 413 K, and even a clear afterglow of 1 s at 433 K, demonstrating excellent HTP properties. In addition, methyl, n-butyl, benzyl, phenyl, and triphenylamine groups (with weak to

strong rotational ability) were further connected to the BCZ molecule to generate five control guests (BCZ–Me, BCZ–nBu, BCZ–Be, BCZ–Ph, and BCZ–TPA). The alkyl groups (such as methyl or n-butyl) have relatively little effect on the excited-state energy of molecules, so the BCZ–Me/PVP and BCZ–nBu/PVP doped materials exhibited almost the same HTP properties as BCZ/PVP. From BCZ–Be to BCZ–TPA, as the rotational ability of the groups increased, the HTP performance of the corresponding doped materials gradually decreased in accordance with the logical and expected trend. Experimental results and theoretical calculations demonstrated that the rigid planar configuration of the guests can resist molecular thermal motion at high temperatures. In addition, the rigid environment provided by the host matrix at high temperatures is an important factor in the HTP properties of the present materials. Next, polyvinyl alcohol (PVA) and polyadiohexylenediamine (PA66) were further selected as the hosts. Although all the doped materials exhibited excellent RTP performance, the HTP performance of doped materials with PVA or PA66 as the host was weaker than that with PVP as the host due to the lower glass transition temperature ($T_g$) values of PVA and PA66 compared to PVP. Finally, the doped materials with four other planar molecules as guests also exhibited excellent HTP properties, indicating that the twofold rigidity strategy has good universality. This work proposes a strategy for constructing organic HTP materials, which will facilitate development of organic phosphorescence materials.

## Results
### Synthesis and photophysical properties
Guests BCZ, BCZ–Me, BCZ–nBu, BCZ–Be, BCZ–Ph, and BCZ–TPA were synthesized in accordance with reported methods (Supplementary Fig. 1; details in Methods). The molecular structures and purities were confirmed by single-crystal X-ray diffraction (XRD), high-resolution mass spectroscopy, nuclear magnetic resonance spectroscopy, and high-performance liquid chromatography (Supplementary Fig. 2,

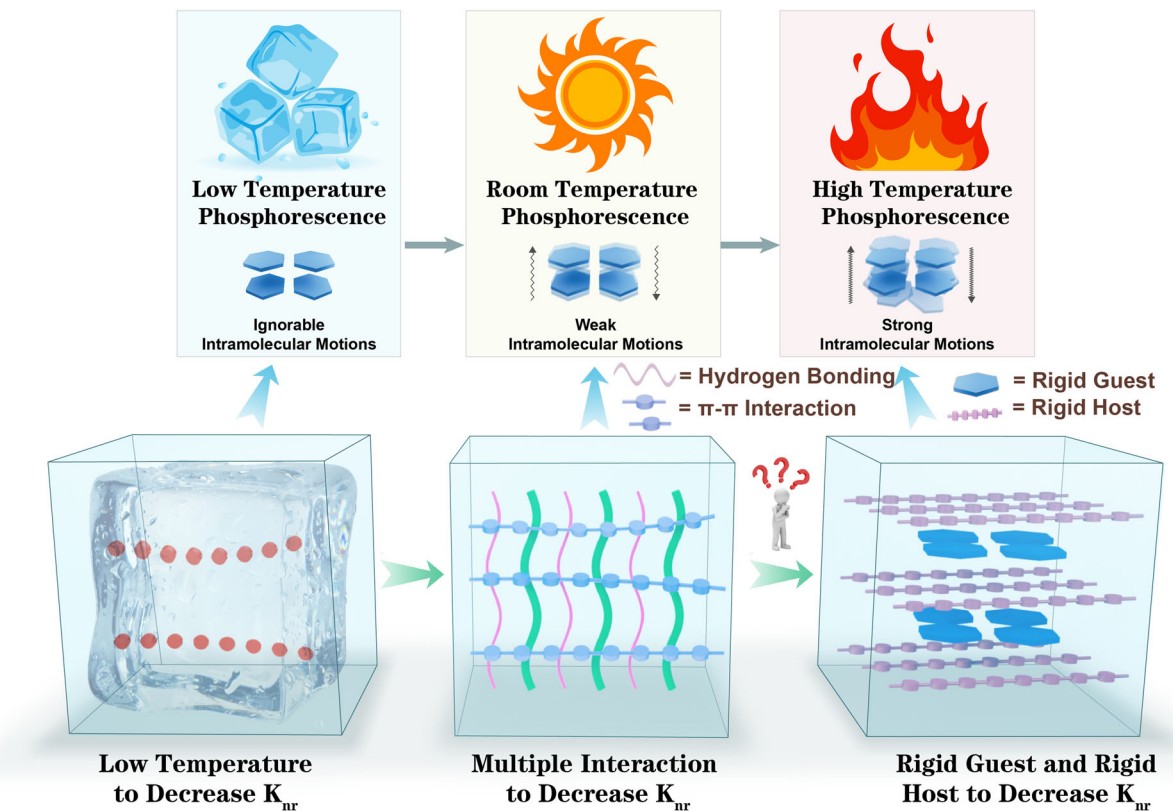

**Fig. 1 | Schematic diagram.** Strategy for synthesizing HTP materials.

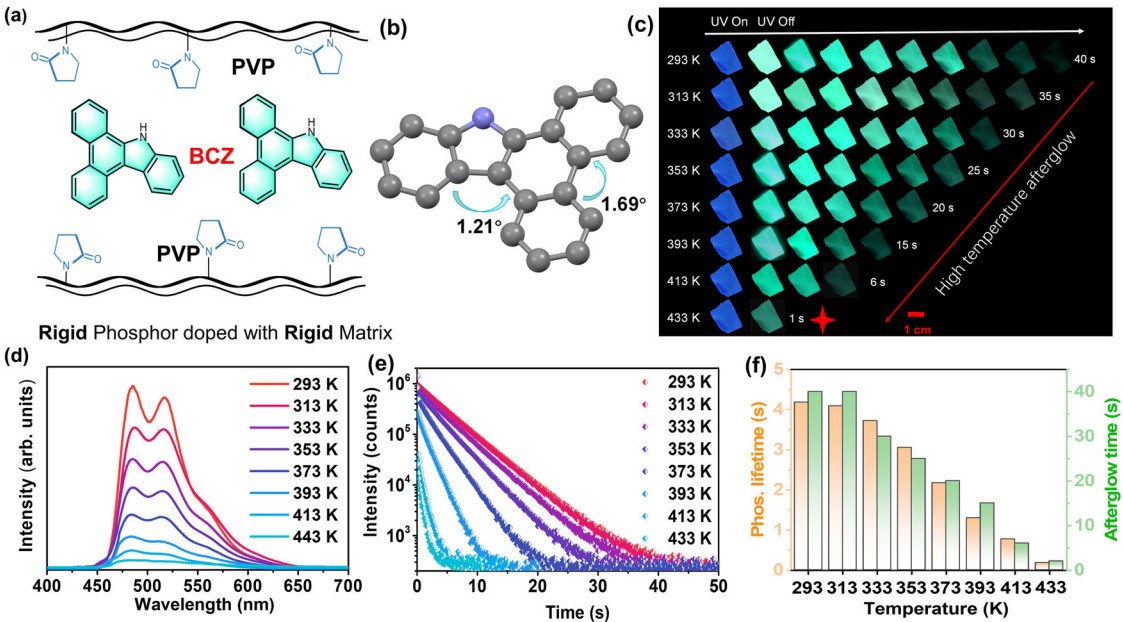

**Fig. 2 | HTP properties of the doped system. a** Molecular structures of the guest BCZ and host PVP. **b** Single-crystal structure of guest BCZ. **c** Afterglow images of doped material BCZ/PVP at various temperatures. **d** Phosphorescence spectra of doped material BCZ/PVP at various temperatures (Excitation wavelength: 380 nm; Delayed time: 1 ms). **e** Kinetic attenuation curves of doped material BCZ/PVP at various temperatures (Excitation wavelength: 380 nm). **f** Phosphorescence lifetime and afterglow time of doped material BCZ/PVP at various temperatures (Excitation wavelength: 380 nm, orange: phosphorescence lifetime, green: afterglow time).

Supplementary Figs. 31–60). The torsion angles of BCZ were only 1.21° and 1.69° (Fig. 2b). The almost completely planar rigid configuration can restrict intramolecular motion and thereby suppress non-radiative transitions. Therefore, BCZ exhibited strong blue fluorescence in the solution state with a fluorescence quantum yield (Q.Y.) of 75% at room temperature (293 K), and a green afterglow of up to 53 s with a 6.36 s lifetime at low temperature (77 K) (Supplementary Figs. 3 and 4). Commercially available PVP was tested by differential scanning calorimetry; the $T_g$ value was about 456 K (Supplementary Fig. 5). The doped materials were fabricated by dissolution–evaporation (details in Methods). A series of BCZ/PVP doped materials with various guest–host mass ratios (0.1:100 to 5:100) were prepared. The doped materials exhibited the strongest phosphorescence emission when the mass ratio was 0.5:100 (Supplementary Fig. 6). Therefore, doped materials with a guest mass percentage of 0.5 wt% were selected as the research targets.

## HTP properties of the doped system

BCZ/PVP was activated after irradiation with a 365 nm ultraviolet lamp for 5 s. Then, the doped materials exhibited strong blue fluorescence with a wavelength of 428 nm and a green afterglow with an emission wavelength of 486/518 nm for about 40 s after termination of the excitation source (Fig. 2c, Supplementary Fig. 7). The delayed emission spectra at various temperatures from 77 K to 293 K indicated that the emission intensity gradually decreased as the temperature increased, confirming that the afterglow was phosphorescence rather than thermally activated delayed fluorescence (Supplementary Fig. 8). The phosphorescence Q.Y. was 22.4% and the lifetime was as long as 4.18 s for BCZ/PVP at room temperature. Next, the focus was on studying the luminescence performance of doped materials at high ambient temperatures. Although the phosphorescence intensity and lifetime of BCZ/PVP decreased at high temperatures (Fig. 2d, e), there was a strong green afterglow of up to 20 s with a 2.17 s lifetime at 373 K. When temperature was increased to 413 K, the afterglow time was satisfactory for 6 s with a 0.77 s lifetime. Even at a high temperature of 433 K, the doped material exhibited a 1 s afterglow that was evident to the unaided eye (Fig. 2c, f). In addition, the phosphorescence Q.Y. of

BCZ/PVP was maintained at 9.2%, 4.7%, and 1.8% at 373 K, 413 K, and 433 K, respectively. After the temperature was decreased to room temperature, the afterglow time and brightness of BCZ/PVP returned to the initial level (Supplementary Fig. 9). Thus, the phosphorescence emission of the doped material exhibited strong high-temperature resistance, and confirmed the feasibility of constructing HTP materials with rigid planar molecules as guests.

## Influence of guests on the HTP

To further verify the feasibility of the rigid planar molecular strategy, methyl (Me), *n*-butyl (*n*Bu), benzyl (Be), phenyl (Ph), and triphenylamine (TPA) groups with weak to strong rotational ability were connected to the BCZ molecule to generate the control guests (BCZ–Me, BCZ–*n*Bu, BCZ–Be, BCZ–Ph, and BCZ–TPA, respectively; Fig. 3a). The control guests did not contain the N–H group, thereby eliminating the possibility of hydrogen bonding with the polymer. The single-crystal results indicated that regardless of the group to which the guest was connected, the configuration change of the dibenzo[*a,c*]carbazole component was small, and the torsion angles were 0.61° to 4.54° and 1.93° to 6.83° (Fig. 3a). However, the torsion angles between the benzene ring and dibenzo[*a,c*]carbazole component of BCZ–Ph and BCZ–TPA were 98.09° and 80.79°, respectively (Fig. 3a). In addition, the optimal molecular configuration obtained by theoretical calculations indicated that the configuration of the dibenzo[*a,c*]carbazole component of all the guests was almost planar, and the torsion angles were 1.2° to 4.1° (Supplementary Fig. 10, Supplementary Data 1). Similarly, the torsion angles between the benzene ring and dibenzo[*a,c*]carbazole component of BCZ–Ph and BCZ–TPA were 73.2° and 81.1°, respectively (Supplementary Fig. 10, Supplementary Data 1). Molecular orbital theory calculations indicated that the lowest unoccupied molecular orbital (LUMO) of BCZ–Me and BCZ–*n*Bu was only located on the dibenzo[*a,c*]carbazole component and not on the alkyl group. Because the phenyl group of BCZ–Be was connected to the dibenzo[*a,c*]carbazole component through a methylene unit, the LUMO was also mainly located in the dibenzo[*a,c*]carbazole component rather than the phenyl group. However, for BCZ–Ph and BCZ–TPA, the LUMO was located on the entire molecule, including the

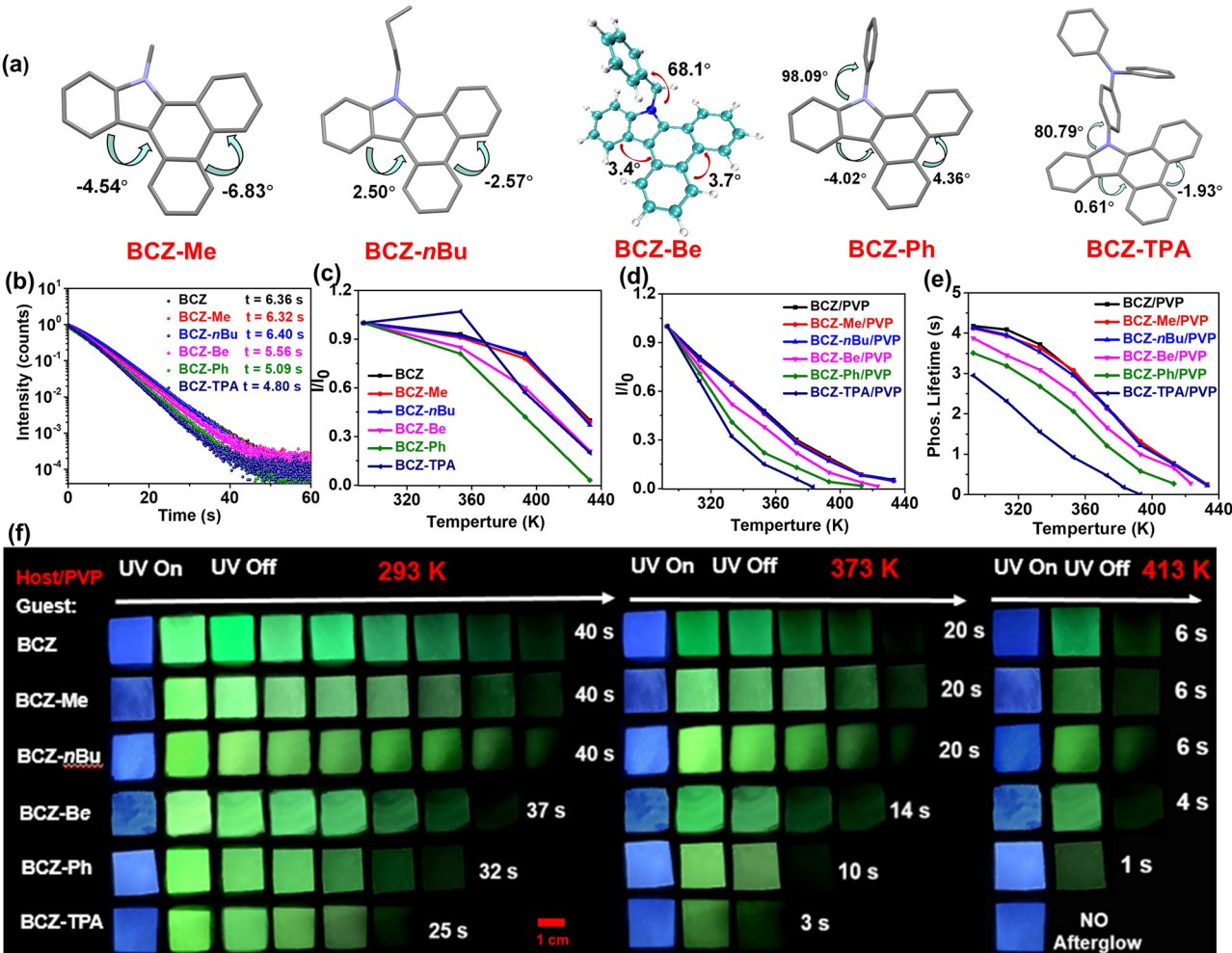

**Fig. 3 | HTP properties of the control doped system. a** Molecular configuration exhibited by single-crystal control guests BCZ–Me, BCZ–nBu, BCZ–Ph, and BCZ–TPA (the molecular configuration of BCZ–Be was simulated through theoretical calculations). **b** Kinetic attenuation curves of six guests at 77 K (Tetrahydrofuran as solvent, $1.0 \times 10^{-5}$ mol/L). **c** Relative changes in fluorescence intensity of six guests at various temperatures (DMSO as solvent, $1.0 \times 10^{-5}$ mol/L). **d** Relative changes in phosphorescence intensity of six doped materials at various temperatures (Excitation wavelength: 380 nm; delayed time: 1 ms). **e** Phosphorescence lifetime of six doped materials at various temperatures. **f** Afterglow images of six doped materials at various temperatures.

benzene ring (Supplementary Fig. 11). These calculation results indicate that the influence of alkyl substituents on the excited-state energy of the guest can be almost negligible, and the benzyl group exhibited a weak influence, whereas phenyl and triphenylamine groups exhibited a strong influence. In other words, these latter two groups can be used as a rotor to consume the energy of the excited state[44]. To verify this hypothesis, the photophysical properties of all the guests were tested. From BCZ to BCZ–nBu and then to BCZ–TPA, the phosphorescence lifetime (77 K) was initially maintained at about 6.3 s, and then gradually decreased to 4.8 s (Fig. 3b). In addition, the low-temperature phosphorescence intensity and room-temperature fluorescence intensity of six guests (from BCZ to BCZ–TPA) gradually decreased (Supplementary Fig. 12). More importantly, as the temperature was increased from 293 K to 433 K, the magnitude of the decrease in the fluorescence intensity of six guests in dimethyl sulfoxide (DMSO) also gradually increased: decreases of 59%, 61%, 60%, 79%, 97%, and 80% (Fig. 3c, Supplementary Fig. 13). Although the reduction of the fluorescence emission of BCZ–TPA was smaller than that of BCZ–Ph, this is because of the strong solvatochromism of BCZ–TPA (Supplementary Fig. 14). The increase in temperature reduces the polarity of DMSO solvent, leading to a blue shift in the emission wavelength and an increase in the emission intensity; the enhancement effect brought about by the solvatochromism decreases the reduction effect caused

by the thermal motion. In the nonpolar solvent *p*-xylene, as the temperature was increased from 293 K to 403 K, the fluorescence intensity of six guests exhibited a consistent decreasing trend: 35%, 37%, 36%, 47%, 61%, and 78% (Supplementary Fig. 15). Thus, rigid planar molecules exhibited strong resistance to thermal motion at high temperatures. Next, five control doped materials (BCZ–Me/PVP, BCZ–nBu/PVP, BCZ–Be/PVP, BCZ–Ph/PVP, and BCZ–TPA/PVP) were prepared and the phosphorescence spectra of the doped materials at various temperatures were tested. As expected, BCZ–Me/PVP and BCZ–nBu/PVP exhibited almost the same HTP performance as BCZ/PVP, with similar luminescence intensity and afterglow time/phosphorescence lifetime at the same temperature (Fig. 3d–f, Supplementary Figs. 16 and 17). Moreover, consistent with the changing trends of the corresponding guests, from BCZ–Be/PVP to BCZ–TPA/PVP, the attenuation amplitude of phosphorescence performance gradually increased with increasing temperature: from 293 K to 393 K, the phosphorescence intensity decreased by 90%, 95%, and 100%; whereas that of BCZ/PVP to BCZ–nBu/PVP only decreased by about 80% (Fig. 3d, Supplementary Fig. 16). Similarly, from 293 K to 393 K, the phosphorescence lifetime of BCZ–Be/PVP to BCZ–TPA/PVP decreased from 3.88 s to 1.00 s, from 3.41 s to 0.58 s, and from 2.85 s to 0 s, respectively; whereas that of BCZ/PVP to BCZ–nBu/PVP only decreased from about 4.18 s to 1.30 s (Fig. 3e, Supplementary Fig. 17). The afterglow time also

correspondingly decreased (Fig. 3f). The aforementioned control experiments demonstrated that failing to inhibit the molecular motion can substantially damage the HTP properties of organic materials, and confirmed that selecting planar rigid molecules as guests with a weak propensity to undergo molecular motion is an effective strategy for constructing HTP materials. The fluorescence emission of six doped films at various temperatures was also tested. The fluorescence emission of BCZ/PVP, BCZ–Me/PVP, and BCZ–$n$Bu/PVP exhibited almost the same high-temperature resistance properties; whereas from BCZ–Be/PVP to BCZ–TPA/PVP, the high-temperature resistance ability of the fluorescence emission gradually decreased (Supplementary Fig. 18). However, because of the superior stability of singlet excitons compared with triplet excitons, the decrease in the fluorescence intensity of the doped films was less than that of the phosphorescence intensity.

## Energy and optical changes of guests at high temperatures

To further evaluate the configuration changes of the guest at various temperatures, the torsion angles and thermodynamic parameters of BCZ and BCZ–TPA were first calculated at various temperatures (Fig. 4, Supplementary Data 2). The torsional angle changes at positions 1 and 2 of the BCZ molecule at 293 K were small: half peak widths of only 8° and 7.5°, respectively, and the angles with the highest probability of occurrence were 1.6° and 2.1°, respectively (Fig. 4b, c). Thus, the degree of molecular motion was low. Even upon gradually increasing the temperature to 433 K, the change in the twist angles at the two positions compared with that at 293 K was very small (Fig. 4b, c), indicating that the molecular configuration of BCZ was highly stable. Therefore, the BCZ molecule exhibited excellent high-temperature luminescence. Regarding guest BCZ–TPA, as the temperature was increased the changes in the torsion angles at positions 1 and 2 were also almost negligible, indicating that the configuration of the dibenzo[$a$,$c$]carbazole component retained high stability (Fig. 4f, Supplementary Fig. 19). However, regarding the twist angle at position 3 with the triphenylamine group, the magnitude of the torsion varied

substantially with increasing temperature. From 293 K to 443 K, the angle with the highest probability of occurrence gradually increased from 70° to 120°, and the range of angle change was increased by 50° (Fig. 4g). Regarding the other four guests (BCZ–Me, BCZ–$n$Bu, BCZ–Be, and BCZ–Ph), theoretical calculations indicated that the torsional angle changed at positions 1 and 2 of the four guest molecules were also small from 293 K to 433 K (Supplementary Fig. 20, Supplementary Data 2). Similarly to BCZ–TPA, regarding the twist angles of BCZ–Be and BCZ–Ph at position 3, the magnitude of the torsion varied substantially with increasing temperature (Supplementary Fig. 20). Thus, with increasing temperature the rotation of the benzene ring increased sharply, which consumes the energy of the excited state, ultimately deteriorating the HTP performance of the materials. In addition, theoretical calculations were conducted on thermodynamic parameters such as the internal energy (U), enthalpy (H), entropy (S), and Gibbs free energy (G) of guest BCZ and BCZ–TPA at various temperatures. At 293 K, the U of BCZ was 173 J, whereas that of BCZ–TPA was 340 J. The U of BCZ–TPA is 1.96 times that of BCZ, which is basically consistent with the 1.94 times increase in the number of atoms (Fig. 4d, h). However, the $T{\times}S$ (=$H-G$) (T: Temperature, ×: multiple sign, −: minus sign) of BCZ–TPA was 47 J, whereas that of BCZ was 30 J. The former is 1.57 times the latter, indicating that the propensity of BCZ–TPA to undergo molecular motion is stronger than that of BCZ at 293 K. When the temperature was increased to 463 K, although the U of BCZ–TPA was still 1.96 times that of BCZ, $T{\times}S$ increased to 1.65 times (Fig. 4d, h); indicating that the propensity of BCZ–TPA to undergo molecular motion increased with increasing temperature. In addition, regarding the guests BCZ–Me, BCZ–$n$Bu, BCZ–Be, and BCZ–Ph, the $T{\times}S$ at 463 K was 75 J, 83 J, 90 J, and 98 J, respectively; and was 1.04 times, 1.15 times, 1.25 times, and 1.36 times that of BCZ, respectively (Supplementary Fig. 21). The aforementioned calculation results further indicated that the connection to groups with strong rotational ability accelerates thermal motion of molecules at high temperatures, which deteriorates the HTP properties of the present materials.

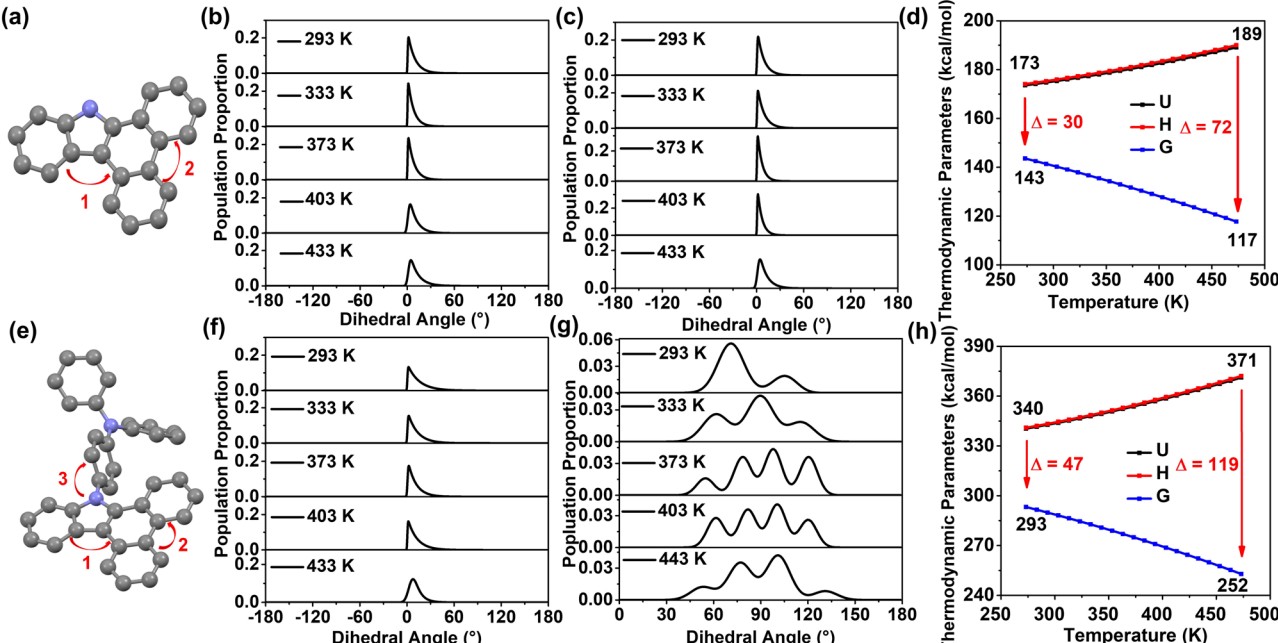

**Fig. 4 | Theoretical calculations of guests at various temperatures. a** Torsion angles of guest BCZ. **b** Distribution of the torsion angle at position 1 of BCZ at various temperatures. **c** Distribution of the torsion angle at position 2 of BCZ at various temperatures. **d** Thermodynamic parameters of guest BCZ at various temperatures. **e** Torsion angles of guest BCZ–TPA. **f** Distribution of the torsion angle at position 1 of BCZ–TPA at various temperatures. **g** Distribution of the torsion angle at position 3 of BCZ–TPA at various temperatures. **h** Thermodynamic parameters of guest BCZ–TPA at various temperatures (U: internal energy, H: enthalpy, G: Gibbs free energy).

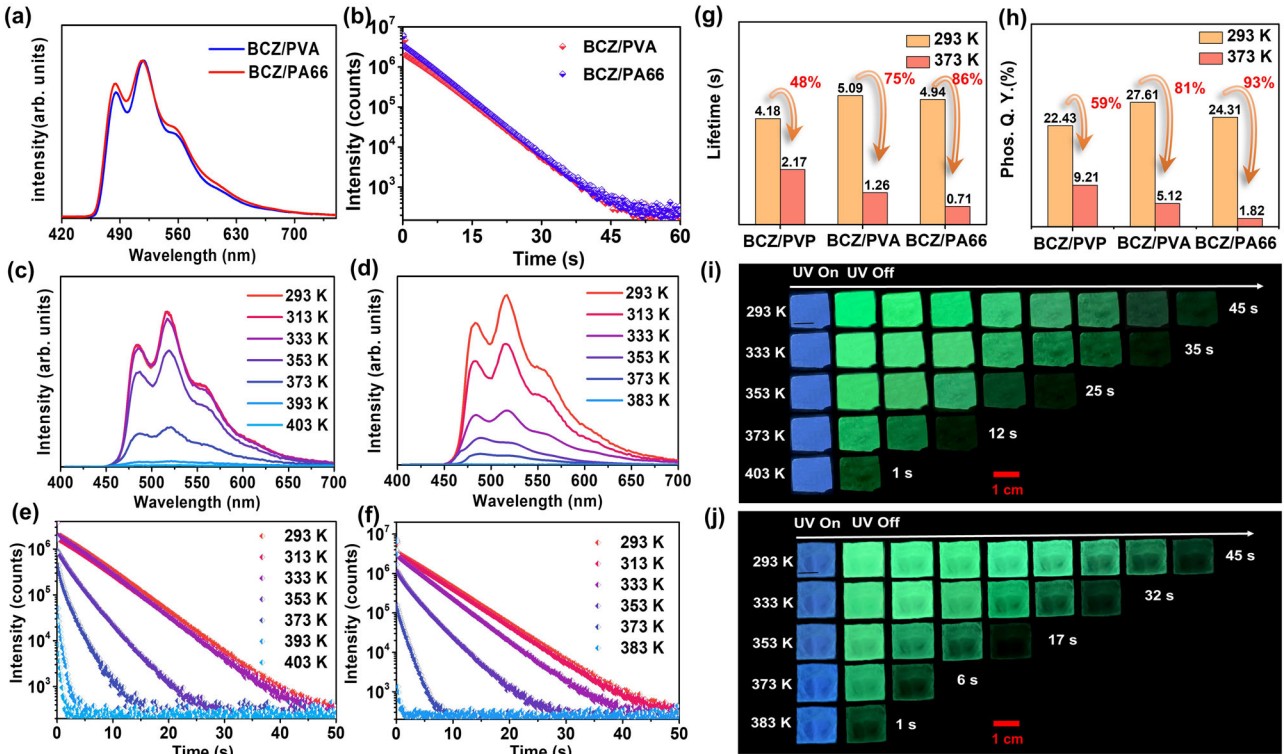

**Fig. 5 | HTP properties of controlled doped systems based on other polymers.**
**a** Phosphorescence spectra of BCZ/PVA and BCZ/PA66 at 293 K (Excitation wavelength: 380 nm; delayed time: 1 ms). **b** Kinetic attenuation curves of BCZ/PVA and BCZ/PA66 at 293 K. Phosphorescence spectra of (**c**) BCZ/PVA and (**d**) BCZ/PA66 at various temperatures (Excitation wavelength: 380 nm; delayed time: 1 ms). Kinetic attenuation curves of (**e**) BCZ/PVA and (**f**) BCZ/PA66 at various temperatures (Excitation wavelength: 380 nm). **g** Phosphorescence lifetime and **h** phosphorescence Q.Y. of BCZ/PVP, BCZ/PVA, and BCZ/PA66 at various temperatures (orange: 293 K, red: 373 K). Afterglow images of **i** BCZ/PVA and **j** BCZ/PA66 at various temperatures.

## Influence of hosts on the HTP

To achieve HTP performance, doped materials not only require a rigid planar configuration of the guest but also a rigid solidification environment provided by the host matrix. Therefore, PVA with a $T_g$ value of 355 K (Supplementary Fig. 22a) and PA66 with a $T_g$ value of 326 K (Supplementary Fig. 22b) were selected as the control hosts for the study. Two control doped materials (BCZ/PVA and BCZ/PA66) were constructed with BCZ as the guest. These two control materials also exhibit an ultralong green afterglow with an emission wavelength of 485/517 nm for about 45 s at 293 K (Fig. 5a). The phosphorescence lifetime was as high as 5.09 s and 4.94 s, respectively (Fig. 5b); and the phosphorescence Q.Y. was 27.6% and 24.3%, respectively. At 293 K, three doped materials (BCZ/PVP, BCZ/PVA, and BCZ/PA66) exhibited almost the same fluorescence wavelength and phosphorescence wavelength (Supplementary Fig. 23), and the difference in the phosphorescence lifetime and phosphorescence Q.Y. was also small. It should be pointed out that from the nonpolar solvent toluene to the polar solvent DMSO, the fluorescence emission of BCZ remained basically unchanged (Supplementary Fig. 24), indicating that guest BCZ had almost no solvatochromism property. Therefore, the influence of various polarities of the hosts on the luminescence performance of doped materials need not be considered. The phosphorescence intensity of BCZ/PVA and BCZ/PA66 decreased sharply after the temperature was increased beyond the $T_g$ (Fig. 5c, d). Moreover, the afterglow duration of BCZ/PVA reduced to 12 s at 373 K and essentially became imperceptible at 403 K (Fig. 5e, i). Whereas, the afterglow of BCZ/PA66 diminished to 6 s at 373 K and was no longer observable at 383 K (Fig. 5f, j). From 293 K to 373 K, the phosphorescence lifetime of BCZ/PVP, BCZ/PVA, and BCZ/PA66 decreased by 48%, 75%, and 86%, respectively (Fig. 5g), and the phosphorescence Q.Y. decreased by 59%, 81%, and 93%, respectively (Fig. 5h). Thus, the

phosphorescence performance of BCZ/PVA and BCZ/PA66 at high temperatures was inferior to that of BCZ/PVP, indicating that host polymers with a low $T_g$ that did not exhibit a rigid solidification ability at high temperatures, deactivating the HTP of corresponding doped materials. However, although the HTP performance decreased, BCZ/PVA and BCZ/PA66 still exhibited a visible afterglow of about 10 s at temperatures 30 K higher than the $T_g$ of the hosts (Fig. 5i, j), which indirectly indicated that the high-temperature resistance of the guest molecules enhances the HTP activity of the materials. Thus, the two-fold rigidity of the host and guest are the two main factors of the HTP activity of the present materials. In addition, XRD of the three polymers was conducted to evaluate the effect of the morphology on the phosphorescence performance of the doped materials, and the results shown that polymers PVP and PVA are amorphous state (Supplementary Fig. 25a, b). Because of the low content of the guest in the doped materials, the morphology of the doped materials remains amorphous state. Although polymer PA66 and doped material BCZ/PA66 exhibited some crystallinity, they were predominant in amorphous form (Supplementary Fig. 25c). Thus, the morphology of the polymers is not the main factor in the HTP properties of the present doped materials.

## Universality of strategy

To verify the universality of this strategy, four rigid planar compounds, such as 7H-benzo[c]carbazole (cBCZ), 7H-dibenzo[c,g]carbazole (cgBCZ), 7-methyl-7H-benzo[c]carbazole (cBCZ–Me), and 7-methyl-7H- dibenzo[c,g]carbazole (cgBCZ–Me), were selected or synthesized as guests to construct doped systems with PVP as the host (Fig. 6a). The molecular structures and purities of four guests were confirmed by single-crystal X-ray diffraction, high-resolution mass spectroscopy, nuclear magnetic resonance spectroscopy, and high-performance liquid chromatography (Supplementary Fig. 26, Supplementary

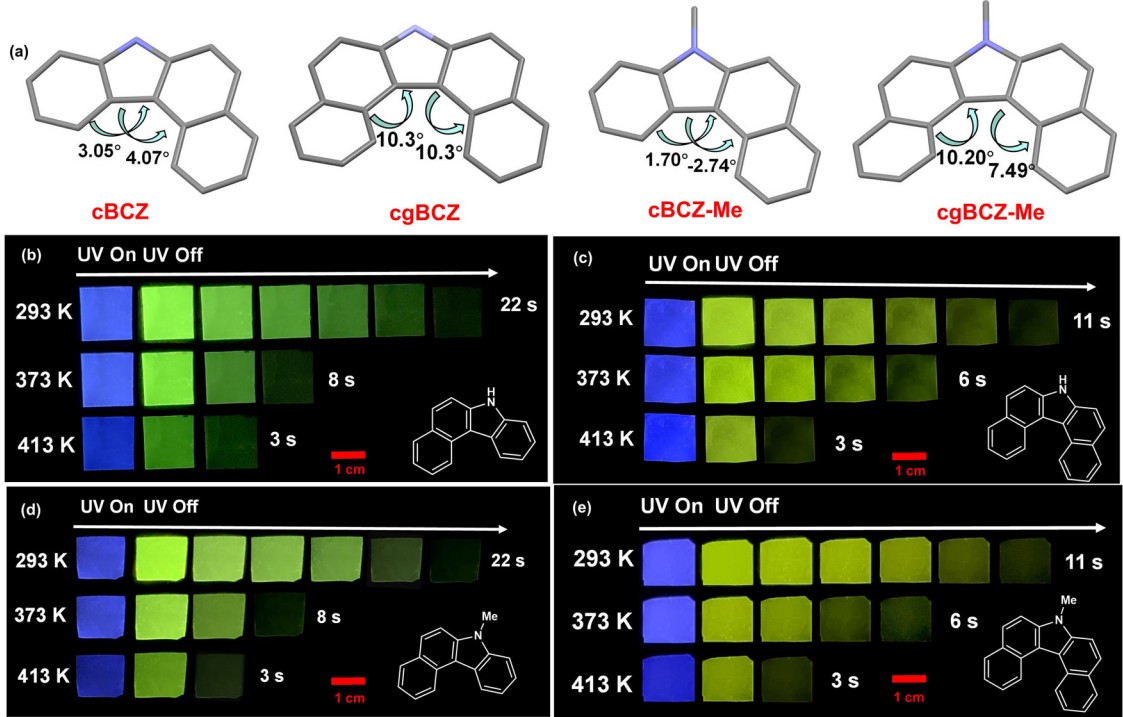

**Fig. 6 | HTP properties of controlled doped systems based on other guests. a** Single-crystal structures of guests. Afterglow images of (**b**) cBCZ/PVP, (**c**) cBCZ-Me/PVP, (**d**) cgBCZ/PVP, and (**e**) cgBCZ−Me/PVP at various temperatures.

Figs. 31–60). All four doped materials exhibited excellent HTP properties (Fig. 6b–e, Supplementary Fig. 27). The afterglow of cBCZ/PVP and cBCZ−Me/PVP was approximately 22 s, 9 s, and 4 s at 293 K, 373 K, and 413 K, respectively (Fig. 6b, c). The phosphorescence lifetime of cBCZ/PVP were 2.32 s, 0.91 s, and 0.40 s at 293 K, 373 K, and 413 K, respectively (Supplementary Fig. 28a), and those of cBCZ-Me/PVP were 2.28 s, 0.88 s, and 0.38 s, respectively (Supplementary Fig. 28b). The phosphorescence Q.Y. values of cBCZ/PVP were 16.4%, 8.2%, and 2.5% at 293 K, 373 K, and 413 K, respectively, and those of cBCZ-Me/PVP were 17.0%, 7.8%, and 2.3%, respectively. The corresponding afterglow times of cgBCZ/PVP and cgBCZ−Me/PVP were approximately 11 s, 6 s, and 2 s at 293 K, 373 K, and 413 K, respectively (Fig. 6d, e). The phosphorescence lifetime of cgBCZ/PVP were 1.36 s, 0.63 s, and 0.21 s at 293 K, 373 K, and 413 K, respectively (Supplementary Fig. 28c), and those of cgBCZ-Me/PVP were 1.40 s, 0.66 s, and 0.23 s, respectively (Supplementary Fig. 28d). The phosphorescence Q.Y. values of cgBCZ/PVP were 14.1%, 6.8%, and 2.0% at 293 K, 373 K, and 413 K, respectively, and those of cgBCZ-Me/PVP were 13.8%, 6.4%, and 2.1%, respectively. Compared to cgBCZ/PVP and cgBCZ−Me/PVP, cBCZ/PVP and cBCZ−Me/PVP have relatively better phosphorescence performance. This is because the torsion angles of cBCZ and cBCZ−Me are 3.05° and 4.07°, 1.70° and 2.74°, respectively (Fig. 6a); whereas the torsion angles of cgBCZ and cgBCZ−Me are 10.30° and 10.30°, 10.20° and 7.49°, respectively (Fig. 6a). The distorted spatial configurations of cgBCZ and cgBCZ−Me increase the degree of molecular motion and reduce the HTP activities of the doped materials. The above results indicate that the strategy for preparing HTP materials by using planar rigid molecules as guests has good universality.

### Application of HTP materials to fire protection

Indoor fires can cause a series of harsh conditions such as high temperatures, thick smoke, and power outages. These conditions reduce visibility, imparting difficulties to firefighters in recognizing each other during disaster relief, leading to injury or death. Therefore, there is an urgent need for personnel identification devices for high-temperature

and thick-smoke environments. The smoke penetration ability of luminescence sources such as fluorescence and phosphorescence are stronger than that of incandescence sources such as common lighting lamps. In addition, phosphorescence materials with a long afterglow time do not require constant excitation, which not only saves electricity but also reduces the radiation of the excitation source to the human body. Therefore, phosphorescence materials are suitable for luminescent recognition in search-and-rescue operations. However, conventional organic phosphorescence materials are sensitive to temperature, and the high temperature generated by fires can substantially quench their phosphorescence emission. Therefore, the long-lifetime and high-temperature resistant phosphorescence material that was designed in this study is expected to help identify firefighters during rescue and firefighting operations. Accordingly, a convenient device was prepared that emits phosphorescence upon straightforward excitation. The device consists of a battery, a controller, three excitation light bulbs (Ex. = 380 nm, $P$ = 10 W), and a HTP film (area: 10 cm × 8 cm), and can be placed on the clothes of firefighters (Fig. 7a, Supplementary Fig. 29). The controller can adjust the switching time of the excitation light bulbs in accordance with the present situation; in a normal environment, the light bulbs excite the thin film for 3 s every 30 s, and the long afterglow time of the HTP film can ensure that the device is continuously in a luminous state (Supplementary Fig. 30). Next, the luminous phenomenon of the device was tested in a high-smoke environment (two smoke grenades were fired in a 60 m³ confined space; the visibility in the space was <1.0 m; the equipment was placed on a rack rather than on the human model); the light bulbs excited the thin film for 3 s every 15 s (Fig. 7b). The focus was on testing the penetration distance of the phosphorescence emitted by the recognition device. At a distance of 1 m from the device, a striking green light plate was clearly visible (Fig. 7c). This brightness was sustained even when the distance was extended to 3 m. At a distance of 5 m, the green afterglow remained distinct. Impressively, even at a distance of 10 m, the green luminescent points were still discernible to the naked eye. Subsequently, the luminescence of the

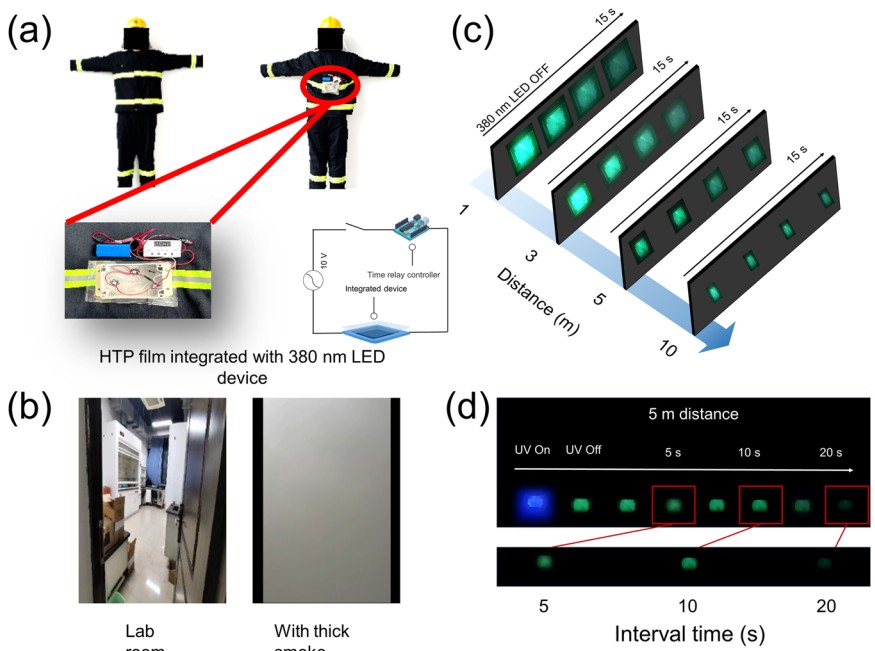

**Fig. 7 | Application of HTP materials. a** Physical and schematic diagrams of the device (Written informed consent had been obtained from the model; LED light emitting diode). **b** Simulated smoke environment. **c** Penetration distance of the device in thick smoke. **d** Luminescence phenomenon of the device in thick smoke at various excitation time intervals.

device was evaluated at different excitation intervals from a distance of 5 m. When the excitation lamp was switched on for 2 s every 5 s, the device consistently exhibited a bright luminous effect throughout the entire duration. Increasing the interval to 10 s, the device continued to display adequate brightness. However, extending the interval to 20 s resulted in a noticeable reduction in afterglow intensity during the final 3 to 5 s, making it challenging to observe the green afterglow without aid (Fig. 7d). Therefore, the interval time of the excitation light must be adjusted in accordance with the present situation. Accordingly, the present device can play a role in identification and guidance in dense-smoke environments. This device can be applied not only in high-temperature and thick-smoke scenarios but also in low-visibility environments (such as in underground coal mines filled with dust, as well as recognition of outdoor cleaners and traffic police, during dawn, night, haze, and heavy fog). This attempt has promoted the application range of organic phosphorescent materials, especially organic HTP materials.

## Discussion

A strategy was established to construct organic doped HTP materials by using planar rigid molecules as guests and polymers with high glass transition temperatures as hosts. The planar rigid structure of the guests resisted molecular thermal motion at high temperatures, thereby reducing non-radiative transitions. The rigid environment provided by the host matrix further restricted the molecular motion of the guest. The guest–host twofold rigidity activated the HTP properties of the doped materials. In addition, the twofold rigidity strategy has good universality. By using the long afterglow and high-temperature resistance advantages of doped material, the organic phosphorescence material was applied to identifying and rescuing firefighters.

## Methods
### Synthesis or purchase of guest compounds
A mixture of indole-2-carboxylic acid (322.2 mg, 2 mmol), cyclic diaryliodonium salt (1027.2 mg, 2.4 mmol), Pd(OAc)$_2$ (44.8 mg, 0.2 mmol), K$_2$CO$_3$ (607.3 mg, 4.4 mmol), and HOAc (12 mL) was stirred

at 145 °C for 12 h. After cooling to the room temperature, the reaction mixture was poured into CH$_2$Cl$_2$ (100 mL); the organic layer was washed with water (50 mL) three times and then dried over Na$_2$SO$_4$. After removing the solvent under reduced pressure, the residue was purified by flash chromatography on silica gel to afford the target product guest BCZ.

A mixture of BCZ (268 mg, 1 mmol), potassium *tert*-butoxide (168 mg, 1.5 mmol), 1-iodomethane (198 mg, 1.4 mmol), and THF (20 mL) was stirred at 50 °C for 2 h. After cooling to room temperature, the reaction mixture was poured into CH$_2$Cl$_2$ (100 mL); the organic layer was washed with water (50 mL) three times and then dried over Na$_2$SO$_4$. After removing the solvent under reduced pressure, the residue was purified by flash chromatography on silica gel to afford the target product guest BCZ–Me.

To a solution of 2-iodoaniline (2.2 g, 10.0 mmol) in triethylamine (30.0 mL), PdCl$_2$(PPh$_3$)$_2$ (70.1 mg, 0.1 mmol), CuI (58.0 mg, 0.3 mmol), and phenylacetylene (1.2 g, 12.0 mmol) were successively added at room temperature under N$_2$. After stirring for 12 h at room temperature, the reaction mixture was poured into CH$_2$Cl$_2$ (200 mL); the organic layer was washed with water (100 mL) three times and then dried over Na$_2$SO$_4$. After removing the solvent under reduced pressure, the residue was purified by flash chromatography on silica gel to afford pure 2-(phenylethynyl)aniline. A mixture of 2-(phenylethynyl)aniline (1.9 g, 10.0 mmol), K$_2$CO$_3$ (4.1 g, 30.0 mmol), and 1-iodobutane (3.3 mL, 30.0 mmol) in 20.0 mL CH$_3$CN was stirred at 125 °C for 48 h. After cooling to room temperature, the reaction mixture was poured into CH$_2$Cl$_2$ (200 mL); the organic layer was washed with water (100 mL) three times and then dried over Na$_2$SO$_4$. After removing the solvent under reduced pressure, the residue was purified by flash chromatography on silica gel to afford pure *N,N*-dibutyl-2-(phenylethynyl)aniline. Then, to a 25 mL Schlenk tube, *N,N*-dibutyl-2-(phenylethynyl)aniline (61.6 mg, 0.2 mmol), *o*-bromobenzoic acid (60.3 mg, 0.3 mmol), Pd(OAc)$_2$ (4.5 mg, 0.02 mmol), PivOK (56.0 mg, 0.4 mmol), Cu(PivO)$_2$ (106 mg, 0.4 mmol), and dimethylacetamide (2.0 mL) were added. The tube was then evacuated briefly under high vacuum, charged with oxygen, and stirred at 130 °C for 18 h. After removing the

solvent under reduced pressure, the residue was purified by flash chromatography on silica gel to afford the target product BCZ−*n*Bu.

NaH (120 mg, 60% dispersion in mineral oil, 3 mmol) was added to a 50 mL flask. Then, BCZ (268 mg, 1 mmol) in *N,N*-dimethylformamide (4 mL) was added dropwise under stirring. The mixture was stirred in an ice bath for 10 min and then stirred at room temperature for 30 min. Afterward a solution of 4-bromobenzyl bromide (205 mg, 1.2 mmol) in *N,N*-dimethylformamide (4 mL) was added dropwise. The reaction mixture was stirred overnight at room temperature. After quenching with water, the reaction mixture was poured into $CH_2Cl_2$ (100 mL); the organic layer was washed with water (50 mL) three times and then dried over $Na_2SO_4$. After removing the solvent under reduced pressure, the residue was purified by flash chromatography on silica gel to afford the target product BCZ−Be.

A mixture of BCZ (268 mg, 1 mmol), iodobenzene or 4-iodotriphenylamine (1.2 mmol), active copper powder (455 mg, 7 mmol), 18-crown-6 (49 mg, 0.2 mmol), $K_2CO_3$ (1133 mg, 8.2 mmol), and 1,2-dichlorobenzene (30 mL) was stirred at 180 °C for 72 h under $N_2$. This was followed by cooling to room temperature and filtration (to remove excess copper powder and inorganic salts). The organic solvent was removed under reduced pressure. The residue was dissolved with dichloromethane (DCM), washed with saturated salt water, extracted with DCM three times, and then dried over $Na_2SO_4$. After removing solvent under reduced pressure, the residue was purified by flash chromatography on silica gel to afford the target product BCZ−Ph or BCZ−TPA.

The cBCZ and cgBCZ were commercially purchased but required further purification through column chromatography, with a purity of 99% at the time of commercial purchase.

A mixture of cBCZ or cgBCZ (1 mmol) and potassium *tert*-butoxide (168 mg, 1.5 mmol) were dissolved in THF at 50 °C, and then iodomethane (198 mg, 1.4 mmol) was added. After being cooled to the room temperature, the reaction mixture was poured into $CH_2Cl_2$ (100 mL). The organic layer was washed with water (50 mL) for three times, and then dried over $Na_2SO_4$. After the removal of solvent under reduced pressure, the residue was purified by flash chromatography on silica gel to afford the target product cBCZ-Me or cgBCZ-Me.

### Preparation of doped materials with PVP as the host
A total of 300 mg of PVP host (molecular weight: 360,000 g/mol, directly use after purchase), 1.5 mg of guest, and 10 mL of DCM were to a 50 mL beaker, and sonicated for 30 min for full dissolution. The resulting mixture was poured into a mold and placed in an oven (not under nitrogen). The material was first baked at 40 °C for 3 h and then baked at 100 °C for 12 h to obtain the final doped materials.

### Preparation of doped materials with PVA as the host
A total of 300 mg of PVA (molecular weight: 85,000 to 124,000 g/mol, directly use after purchase) and 10 mL of distilled water were added to a 50 mL beaker, and heated at 90 °C for 10 h for full dissolution. One milliliter of DMSO was added to 1.5 mg of the guest; then this DMSO solution was added to the previously prepared aqueous solution. The resulting mixture was placed into a mold and placed in an oven (not under nitrogen). The material was first baked at 80 °C for 3 h and then baked at 100 °C for 12 h to obtain the final doped materials.

### Preparation method of doped materials with PA66 as the host
A total of 300 mg of PA66 (molecular weight: 43000 g/mol, directly use after purchase) and 10 mL of formic acid were added to a 50 mL beaker, and heated at 90 °C for 10 h for full dissolution. One milliliter of DMSO was added to 1.5 mg of the guest; then this DMSO solution was added to the previously prepared formic acid solution. The resulting mixture was placed into a mold and placed in an oven (not under nitrogen). The material was first baked at 80 °C for 3 h and then baked at 100 °C for 12 h to obtain the final doped materials.

**Theoretical calculation methods.** Geometry optimizations and frequency analyses were conducted at M06-2X/6-311 g(d) level in the gas phase within Gaussian 09[45]. An accurate DLPNO-CCSD(T) method with tightPNO and tightSCF settings in combination with the cc-pVTZ basis set[46] was used for evaluating the electronic energy component of the free energies via ORCA 5.0.1 code[46–48]. The thermal correction to the free energies was calculated with Shermo code based on Grimme's quasi-rigid-rotor harmonic oscillator model[49,50]. ZPE and $[H_{298} - H_0]$ corrections were calculated by using vibrational frequencies scaled by 0.977 at the level of B3LYP/6-31 G(d)[51]. Calculations of single-molecule dynamics were carried out with ORCA 5.0.1 code at the level of B97-3c to obtain the molecular conformations of the guests at four temperatures[52,53]. Four-picosecond molecular dynamics (MD) simulations were performed with a time step of 1 fs at temperatures of 293 K, 333 K, 373 K, 403 K, and 443 K. The temperature was controlled with the canonical sampling through velocity rescaling (CSVR) thermostat. Each system was equilibrated for 0.5 ps and the conformations were stored at each time step during the last 3.5 ps of MD simulations to obtain dihedral angle populations.

### Reporting summary
Further information on research design is available in the Nature Portfolio Reporting Summary linked to this article.

### Data availability
All the data supporting the findings in this work are available within the manuscript and Supplementary Information file and available from the corresponding authors upon request. The source data generated in this study have been deposited in the Figshare database (https://doi.org/10.6084/m9.figshare.25051373). The X-ray crystallographic coordinates for structures reported in this study have been deposited at the Cambridge Crystallographic Data Centre, under deposition numbers 2254075, 1919915, 2264139, 2254077, 2263842, 1412957, 2098803, 2255662, and 2255665 of BCZ, BCZ−Me, BCZ−*n*Bu, BCZ−Ph, BCZ−TPA, cBCZ, cgBCZ, cBCZ-Me, and cgBCZ-Me, respectively (www.ccdc.cam.ac.uk/data_request/cif). Source data are provided with this paper.

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

## Acknowledgements

This work was supported by the financial support from the National Natural Science Foundation of China (No. 22105148, received by Y.L.; No. 22071184, received by X.H.; No. 22125803, and 22020102006, received by X.M.), the Zhejiang Provincial Natural Science Foundation of China (No. LY20B020014, received by X.H.), and Xinmiao Foundation of Zhejiang Province (No. 2023R451051, received by K.C.). Thank the Security Office of Wenzhou University and the Special Police Brigade of Wenzhou Fire Rescue Detachment for providing fire-fighting equipment and technical guidance.

## Author contributions

Y. L, X. H. and X. M. designed the research work and revised the manuscipt. K. C. synthesized and prepared the materials. K. C. and Y. Z. carried out photophysical property measurements. Y. Z. carried out density functional theory calculations. Y. L, and X. H. wrote the manuscript. W. D, M. L, Z. C. and H. W. edited the manuscript. All authors discussed the results and commented on the manuscript.

## Competing interests

The authors declare no competing interests.
