## [Peer Review File · Nature Communications]

Twofold Rigidity Activates Ultralong Organic High-Temperature PhosphorescenceREVIEWER COMMENTS

Reviewer #1 (Remarks to the Author):

In this paper, Ma et al. first cleverly selected a compound with almost completely planar molecular configuration as guest, and then chose the polymer with high glass transition temperature as host. The phosphorescence emission intensity of the constructed doped system exhibits good high-temperature resistance. Subsequently, the authors designed and synthesized a series of compounds with different movement ability as control guests, and the experimental results demonstrated that the stronger the thermal motion of molecules at high temperatures, the weaker the high-temperature phosphorescence performance of the doped materials. The biggest contribution of this work is the first clear proposal of a method for designing high-temperature phosphorescence materials, and the authors have demonstrated in the manuscript that this method has good universality. Finally, the authors attempted to use this type of material for the identification of personnel in fires. Although this application currently appears to require overcoming many problems, this attempt has expanded the potential application range of organic phosphorescence materials, which is worthy of recognition. In summary, the dual rigidity strategy provided in this manuscript is an effective option for designing high-temperature resistant phosphorescence materials, which is of important significance for promoting the development of the entire organic phosphorescence materials. Therefore, this reviewer suggest that the manuscript can be published in the Nature Communications after minor revisions.

1. Although the authors provided the single crystal structure and NMR spectra of the guest molecules, this reviewer still suggest that the authors should provide high-resolution mass spectra of the guests.
2. Due to the significant differences in solubility of different polymers in different solvents, the authors should provide a detailed description of the preparation process of doped thin films.
3. Although this manuscript mainly studies the phosphorescence performance of film material at different temperatures, this reviewer is still curious about whether the fluorescence emission of film material also has high-temperature resistance property.
4. The annotation of Fig. 3b is incomplete and the authors need to supplement it.
5. The characters in all the figures are difficult to perceive clearly. The quality of the figures should be improved.
6. In the section of the "Influence of hosts on the Property of HTP", the authors seem to have omitted the description of the subject PVA.
7. The authors need to provide the specific dimensions of the luminescent film in the display device.

Reviewer #2 (Remarks to the Author):

In this manuscript, the authors proposed a strategy to obtain high temperature phosphorescence (HTP) materials by choosing planar rigid molecule as guest, and rigid polymer as host matrix to suppress the nonradiative transition resulting from the thermal vibration at high temperatures. The twofold rigidity of

doped systems based on guests with the planar rigid configuration and host polymers with high glass transition temperatures was the key to achieve ultralong HTP property. The HTP performance of the doped materials gradually decreased with the increased rotational ability of the connected group changing from methyl to triphenylamine. The influence of guests and theoretical calculations about energy and optical changes at high temperatures were provided. According to the HTP performance with ultralong afterglow and high temperature resistance, the identification and rescue of firefighters was attempted by utilizing doped materials. However, a comprehensive analysis based on experimental results and theoretical calculations was not provided to get insight into the mechanism of HTP behavior on doped materials, while similar HTP behavior has been reported. In addition, the writing of the manuscript should be improved. Numerous mistakes were found, indicating that the manuscript was poorly prepared.

1. The manuscript was written in an extremely Chinglish style, with inconsistent verb tenses and other grammatical mistakes. For example, "As show in Fig. 2b..." (Page 4, line 129) should be "As shown in Fig. 2b...". "and the differential scanning calorimetry result show..." (Page 4, line 136) should be "and the differential scanning calorimetry result shows...". Many similar mistakes could be found. In a word, the manuscript should be rewritten.
2. The abbreviations in the manuscript were provided following the full name with a slash (polyvinyl pyrrolidone/PVP, polyvinyl alcohol/PVA, Tetrahydrofuran/THF, high glass transition temperatures/Tg, internal energy/U, enthalpy/H, entropy/S, and Gibbs free energy/G), while the names of doped materials were also provided in the same form (BCZ/PVP). It was confused, and the abbreviation should be provided in the correct form.
3. The format errors in the manuscript should be carefully checked. For example, the space should not be omitted. "Fig.3" (Page 4, line 135,); "Fig.5" (Page 4, line 137,); "50mL" (Page 15, line 467). In addition, the name of "PVA" should be added after "Therefore" on line 296 of page 10. The word "Figure 6a" should be updated as "Fig. 6a" (Page 11, line 355). The word "the" should be corrected by "The" (Page 15, line 449). Overall, there are so many formatting problems in the manuscript.
4. The authors used single crystal results to study the molecular configuration and torsion angles. However, the doped materials only showed mass ratio of guest molecules of 0.5 wt%, indicating that guest molecules should be isolated in polymer. Thus, the molecular configuration and torsion angles should be different from those in single crystals.
5. The phosphorescence intensity of BCZ and the control guests in solution at 77 K should be provided, which can directly demonstrate that the strong motion ability are not conducive to the phosphorescence performance of guest molecules.
6. The authors compared the liquid fluorescence quantum yields of six guests at room temperature, but only compared the fluorescence intensity at different temperatures. The different sets of criteria could make it confused to study the mechanism.
7. Since the solvent effect had a great influence on the photophysical properties of the guest molecules, it is better to choose a non-polar solvent to study the luminescence performance of guest molecules in solution.
8. The glass transition temperature is important in the manuscript to study the mechanism. Thus, differential scanning calorimetry results and glass transition temperature of all polymers should be provided.
9. The change of hosts polarity could also have influence on the property of HTP as found in solution. A

discussion should be provided.

10. In theoretical calculation, only data of BCZ and BCZ-TPA were presented. What about other molecules?

11. The corresponding names of compounds in Supplementary Fig. 12 should be provided.

12. The doped materials were used to identify rescue workers and trapped personnel in fires, but the cyclicity of the HTP behavior was poor (Supplementary Fig. 9). Are there any related experiments on circulation in fire or smoke environment?

Manuscript ID: NCOMMS-23-33760

TITLE: Twofold Rigidity Activates Ultra-Long Organic High Temperature Phosphorescence

The point-to-point answers to the reviewers' questions are as follows.

Reviewer #1 (Remarks to the Author):

In this paper, Ma et al. first cleverly selected a compound with almost completely planar molecular configuration as guest, and then chose the polymer with high glass transition temperature as host. The phosphorescence emission intensity of the constructed doped system exhibits good high-temperature resistance. Subsequently, the authors designed and synthesized a series of compounds with different movement ability as control guests, and the experimental results demonstrated that the stronger the thermal motion of molecules at high temperatures, the weaker the high-temperature phosphorescence performance of the doped materials. The biggest contribution of this work is the first clear proposal of a method for designing high-temperature phosphorescence materials, and the authors have demonstrated in the manuscript that this method has good universality. Finally, the authors attempted to use this type of material for the identification of personnel in fires. Although this application currently appears to require overcoming many problems, this attempt has expanded the potential application range of organic phosphorescence materials, which is worthy of recognition. In summary, the dual rigidity strategy provided in this manuscript is an effective option for designing high-temperature resistant phosphorescence materials, which is of important significance for promoting the development of the entire organic phosphorescence materials. Therefore, this reviewer suggests that the manuscript can be published in the Nature Communications after minor revisions.

Response: Thanks to the reviewer for his/her evaluation of our work, we have strictly followed the reviewer's suggestions to make revision to the manuscript.

Q1. Although the authors provided the single crystal structure and NMR spectra of the guest molecules, this reviewer still suggest that the authors should provide high-resolution mass spectra of the guests.

Response: Thanks to the reviewer for his/her suggestion, we have supplemented the high-resolution mass spectra of six guests.

(a)

(b)

(c)

Fig. R1. High-resolution mass spectra of six guests (a, **BCZ**; b, **BCZ-Me**; c, **BCZ-*n*Bu**; d, **BCZ-Be**; e, **BCZ-Ph**; f, **BCZ-TPA**).

Q2. Due to the significant differences in solubility of different polymers in different

solvents, the authors should provide a detailed description of the preparation process of doped thin films.

Response: Thanks to the reviewer for his/her suggestion, we have added the detailed process for preparing doped films in the manuscript.

Added:

Preparation of doped materials with PVP as the host

A total of 300 mg of **PVP** host, 1.5 mg of guest, and 10 mL of DCM were to a 50-mL beaker, and sonicated for 30 min for full dissolution. The resulting mixture was poured into a mold and placed in an oven (not under nitrogen). The material was first baked at 40°C for 3 h and then baked at 100°C for 12 h to obtain the final doped materials.

Preparation of doped materials with PVA as the host

A total of 300 mg of **PVA** and 10 mL of distilled water were added to a 50-mL beaker, and heated at 90°C for 10 h for full dissolution. One milliliter of DMSO was added to 1.5 mg of the guest; then this DMSO solution was added to the previously prepared aqueous solution. The resulting mixture was placed into a mold and placed in an oven (not under nitrogen). The material was first baked at 80°C for 3 h and then baked at 100°C for 12 h to obtain the final doped materials.

Preparation method of doped materials with PA6 as the host

A total of 300 mg of **PA6** and 10 mL of formic acid were added to a 50-mL beaker, and heated at 90°C for 10 h for full dissolution. One milliliter of DMSO was added to 1.5 mg of the guest; then this DMSO solution was added to the previously prepared formic acid solution. The resulting mixture was placed into a mold and placed in an oven (not under nitrogen). The material was first baked at 80°C for 3 h and then baked at 100°C for 12 h to obtain the final doped materials.

Q3. Although this manuscript mainly studies the phosphorescence performance of film material at different temperatures, this reviewer is still curious about whether the fluorescence emission of film material also has high-temperature resistance property.

Response: Thanks to the reviewer for his/her suggestion. We tested the fluorescence emission intensity of six doped films at different temperatures. The results show that the fluorescence emissions of **BCZ/PVP**, **BCZ-Me/PVP**, and **BCZ-nBu/PVP** exhibit almost the same high-temperature resistance property, while the high-temperature resistance performance of the fluorescence emission gradually decreasing from **BCZ-Be/PVP** to **BCZ-Ph/PVP** then to **BCZ-TPA/PVP** (Fig. R2). However, due to the superior stability of singlet excitons compared to triplet excitons, the decrease degree in fluorescence emission intensity of doped materials is lower than that of phosphorescence emission. We have added the above results to the revised manuscript.

Fig. R2. Fluorescence emission intensity of six doped films **BCZ-Me/PVP** (a), **BCZ-*n*Bu/PVP** (b), **BCZ-Be/PVP** (d), **BCZ-Ph/PVP** (e), **BCZ-TPA/PVP** (f) at different temperatures.

Q4. The annotation of Fig. 3b is incomplete and the authors need to supplement it.

Response: Thanks to the reviewer for carefully pointing out the errors in the manuscript. We have made corrections in the revised manuscript.

Added: Kinetic attenuation curves of six guests at 77 K (THF as solvent, 1.0×10^{-5} mol/L).

Q5. The characters in all the figures are difficult to perceive clearly. The quality of the figures should be improved.

Response: Thanks to the reviewer for his/her suggestion, we have remade the figures.

Q6. In the section of the "Influence of hosts on the Property of HTP", the authors seem to have omitted the description of the host **PVA**.

Response: Thanks to the reviewer for carefully pointing out the errors in the manuscript. We have added the description of the **PVA** host.

Added: Therefore, polyvinylalcohol (**PVA**) with T_g of 355 K and polyadiahexylenediamine (**PA6**) with T_g of 326 K are selected as the control hosts for the study.

Q7. The authors need to provide the specific dimensions of the luminescent film in the display device.

Response: Thanks to the reviewer for his/her suggestion, we have provided the specific dimensions of the luminescent film in the display device.

Added: Area: 10 cm × 8 cm

Reviewer #2 (Remarks to the Author):

In this manuscript, the authors proposed a strategy to obtain high temperature phosphorescence (HTP) materials by choosing planar rigid molecule as guest, and rigid polymer as host matrix to suppress the non-radiative transition resulting from the thermal vibration at high temperatures. The twofold rigidity of doped systems based on guests with the planar rigid configuration and host polymers with high glass transition temperatures was the key to achieve ultralong HTP property. The HTP performance of the doped materials gradually decreased with the increased rotational ability of the connected group changing from methyl to triphenylamine. The influence of guests and theoretical calculations about energy and optical changes at high temperatures were provided. According to the HTP performance with ultralong afterglow and high temperature resistance, the identification and rescue of firefighters was attempted by utilizing doped materials. However, a comprehensive analysis based on experimental results and theoretical calculations was not provided to get insight into the mechanism of HTP behavior on doped materials, while similar HTP behavior has been reported. In addition, the writing of the manuscript should be improved. Numerous mistakes were found, indicating that the manuscript was poorly prepared.

Response: We are very grateful for the careful review of our work by this reviewer, who provided many valuable suggestions and pointed out the writing of the manuscript. We have made very serious and comprehensive revisions to the writing of the manuscript under the guidance of professionals who use English as their written language. Although there have been significant changes in the English writing of this manuscript, the meaning has not changed significantly. In order not to affect reading, the changes have not been pointed out one by one in the revised manuscript.

As the reviewer mentioned, similar HTP works have indeed been reported, and in fact, we have also provided a detailed explanation in the **Introduction section** of the manuscript. However, these works mainly focus on a series of phosphorescence properties of organic materials at room temperature, only simply describing the phosphorescence phenomenon of materials at different temperatures. There is no systematic and comprehensive study on the HTP properties of the materials, let alone a clear strategy for constructing organic HTP materials. Our work not only constructed organic phosphorescence materials with HTP properties, but also clearly proposed an effective method for preparing HTP materials for the first time, and this method has good universality. In addition, this work also provides a reference for designing HTP materials with special functionality. For example: (i) introducing chiral groups into planar rigid guest molecules, the resulting doped materials may not only exhibit HTP

properties but also exhibit high-temperature circularly polarized emission phenomena; (ii) introducing groups with specific responses to drugs or explosives into rigid planar molecules, the resulting doped materials can recognize the target substance at high temperatures (the above work is already underway). Therefore, this work has certain significance for the development of organic phosphorescence materials. Finally, all authors of this manuscript are very grateful to this reviewer for his/her evaluation and suggestions on the work, which have been very helpful in improving the quality of the manuscript.

Q1. The manuscript was written in an extremely Chinglish style, with inconsistent verb tenses and other grammatical mistakes. For example, “As show in Fig. 2b...” (Page 4, line 129) should be “As shown in Fig. 2b...”. “and the differential scanning calorimetry result show...” (Page 4, line 136) should be “and the differential scanning calorimetry result shows...”. Many similar mistakes could be found. In a word, the manuscript should be rewritten.

Response: Thanks very much to the reviewer for his/her careful review of the manuscript. **We have made very serious and comprehensive revisions to the writing of the manuscript under the guidance of professionals who use English as their written language. Although there have been significant changes in the English writing of this manuscript, the meaning has not changed significantly. In order not to affect reading, the changes have not been pointed out one by one in the revised manuscript.**

Q2. The abbreviations in the manuscript were provided following the full name with a slash (polyvinyl pyrrolidone/PVP, polyvinyl alcohol/PVA, Tetrahydrofuran/THF, high glass transition temperatures/Tg, internal energy/U, enthalpy/H, entropy/S, and Gibbs free energy/G), while the names of doped materials were also provided in the same form (BCZ/PVP). It was confused, and the abbreviation should be provided in the correct form.

Response: Thanks very much to the reviewer for his/her careful review of the manuscript. We have changed the abbreviations in the revised manuscript from the slash to parentheses.

For example: polyvinyl pyrrolidone (PVP), high glass transition temperatures (T_g), internal energy(U), etc.

Q3. The format errors in the manuscript should be carefully checked. For example, the space should not be omitted. “Fig.3” (Page 4, line 135,); “Fig.5” (Page 4, line 137,); “50mL” (Page 15, line 467). In addition, the name of “PVA” should be added after

“Therefore” on line 296 of page 10. The word “Figure 6a” should be updated as “Fig. 6a” (Page 11, line 355). The word “the” should be corrected by “The” (Page 15, line 449). Overall, there are so many formatting problems in the manuscript.

Response: Thanks very much to the reviewer for his/her careful review of the manuscript. We have thoroughly checked and corrected any formatting errors in the revised manuscript.

Q4. The authors used single crystal results to study the molecular configuration and torsion angles. However, the doped materials only showed mass ratio of guest molecules of 0.5 wt%, indicating that guest molecules should be isolated in polymer. Thus, the molecular configuration and torsion angles should be different from those in single crystals.

Response: Thanks to the reviewer for his/her suggestion, we theoretically calculated the optimal configurations of six guest molecules in the gas phase state. The results show that the configuration of the dibenzo[*a,c*]carbazole part is almost planar, the torsion angles are 1.2° - 4.1° (Fig. R3), and the torsion angles between benzene ring and dibenzo[*a,c*]carbazole portion of **BCZ-Ph** and **BCZ-TPA** are 73.2° and 81.1°, respectively (Fig. R3). The molecular configurations calculated theoretically are in good agreement with the configurations displayed by the single crystal structures. We have added the theoretical molecular configuration in the revised manuscript.

Fig. R3. Theoretical molecular configuration of six guests (a, **BCZ**; b, **BCZ-Me**; c, **BCZ-nBu**; d, **BCZ-Be**; e, **BCZ-Ph**; f, **BCZ-TPA**).

Q5. The phosphorescence intensity of **BCZ** and the control guests in solution at 77

K should be provided, which can demonstrate that the strong motion ability are not conducive to the phosphorescence performance of guest molecules.

Response: Thanks to the reviewer for his/her suggestion. We supplemented the low-temperature phosphorescence intensity of six guests under the same conditions. As expected by the reviewer, from **BCZ** to **BCZ-TPA**, the phosphorescence intensity of the guests gradually decreases (Fig. R4), indicating that molecular motion is not conducive to their phosphorescence emission. The above results have been added to the revised manuscript.

Fig. R4. Phosphorescence spectra of six guests at solution state (THF as solvent, 1.0×10^{-5} mol/L, Excitation wavelength: 360 nm).

Q6. The authors compared the liquid fluorescence quantum yields of six guests at room temperature, but only compared the fluorescence intensity at different temperatures. The different sets of criteria could make it confused to study the mechanism.

Response: Thanks to the reviewer for his/her suggestion. As is well known, the intensity of molecular motion directly affects its luminescence intensity. The stronger the motion ability of the connected groups in a molecule, the weaker its luminescence ability. Correspondingly, the more planar and rigid molecular configurations, the stronger the luminescent ability of molecules. Therefore, we plan to demonstrate the impact of guest molecules motion on luminescence performance by comparing the fluorescence intensities of guests at room temperature and the reduction degrees in fluorescence intensity of guests at high temperatures. We have added the fluorescence intensities of the guests at room temperature and removed their fluorescence quantum yields in the revised manuscript.

Added: From **BCZ** to **BCZ-TPA**, the room temperature fluorescence intensity of six guests gradually decreased at room temperature (Fig. R5). More importantly, as the

temperature increased from 293 K to 433 K, the magnitude of the decrease in fluorescence intensity of the six guest molecules also gradually increased, with decreases of 59 %, 61%, 60%, 79%, 97%, and 80%, respectively.

(It must be pointed out that the reduction degree in fluorescence emission of **BCZ-TPA** is less than that of **BCZ-Ph**, which is due to the reverse effect caused by the strong solvatochromism of **BCZ-TPA**, which has been confirmed in testing of non-polar solvent *p*-xylene. For detailed analysis, please refer to the answer to **Q7**).

Fig. R5. Fluorescence emission intensities of six guests at solution state (THF as solvent, 1.0×10^{-5} mol/L, Excitation wavelength: 340 nm).

Q7. Since the solvent effect had a great influence on the photophysical properties of the guest molecules, it is better to choose a non-polar solvent to study the luminescence performance of guest molecules in solution.

Response: Thanks to the reviewer for his/her suggestion. We chose *p*-xylene with low polarity and high boiling point as the solvent. As the boiling point of *p*-xylene is 138°C, the limit for temperature increase is 130°C. Due to the absence of solvatochromism effect interference, the decrease in fluorescence intensity of **CBZ-TPA** is indeed the largest.

Added: In non-polar solvent *p*-xylene, as the temperature increased from 293 K to 403 K, the fluorescence intensity of six guests showed a consistent decreasing trend, decreasing by 19%, 18%, 21%, 34%, 48%, and 66%, respectively (Fig. R6).

Fig. R6. Fluorescence emission spectra of six guests (a, **BCZ**; b, **BCZ-Me**; c, **BCZ-nBu**; d, **BCZ-Bc**; e, **BCZ-Ph**; f, **BCZ-TPA**) at different temperatures (*p*-xylene as solvent, 1.0×10^{-5} mol/L, Excitation wavelength: 340 nm).

Q8. The glass transition temperature is important in the manuscript to study the mechanism. Thus, differential scanning calorimetry results and glass transition temperature of all polymers should be provided.

Response: Thanks to the reviewer for his/her suggestion. We tested the DSC curves of three polymers (**PVP**, **PVA**, and **PA6**), and the results shown that the T_g values of the three polymers are 456 K, 355 K, and 326 K, respectively (Fig, R7). The movement of molecules at high temperatures is the main factor contributing to the deactivation of phosphorescence emissions, and thus designing materials with low molecular motion at high temperatures is an important strategy for achieving high-temperature phosphorescence.

In the host-guest doped system, the luminescent ability of the guest directly determines the luminescent performance of the doped material, while the host plays the important auxiliary role, such as inhibiting the motion of guest molecules and isolating water and oxygen. Based on the above principles, we first choose rigid planar molecule with low molecular motion as the guest, and the more rigid the molecular configuration, the better its high-temperature resistance property. For the host, its primary task is to suppress the motion of guest molecules, especially in high-temperature environments. As is well known, when the external temperature is higher than the T_g of the polymer, the deformation of the polymer will sharply increase, leading to the rapid disappearance of its ability to solidify and inhibit guest molecules. Therefore, the higher T_g of the host, the more favorable the HTP performance of the doped material. Therefore, although the three doped materials (**CBZ/PVP**, **CBZ/PVA**, and **CBZ/PA6**) exhibit similar phosphorescence properties at room temperature, the HTP performance of the doped

materials gradually decreases as the T_g of three polymers gradually decreases.

Fig. R7. Differential scanning calorimetry spectra of PVP (a), PVA (b), and PA6 (c).

Q9. The change of host polarity could also have influence on the property of HTP as found in solution. A discussion should be provided.

Response: Thanks to the reviewer for his/her suggestion. We first tested the fluorescence emission of guest **BCZ** in different solvents. As shown in the Fig. R8, from the non-polar solvent toluene to the polar solvent DMSO, the changes in fluorescence wavelength and fluorescence intensity of **BCZ** can be ignored and have no regularity, indicating that **BCZ** has no solvatochromic property. This is because there is no electron-donating or electron-withdrawing group in the **BCZ** molecule (not D-A or D- π -A type molecule), so it is difficult to form intramolecular charge transfer. Therefore, the fluorescence emission wavelengths (Fig. R9a), phosphorescence emission wavelengths (Fig. R9b), phosphorescence lifetimes, and phosphorescence Q.Y. values of **BCZ/PVP**, **BCZ/PVA**, and **BCZ/PA6** are also basically the same. Therefore, the polarity of the hosts is not the main reason for affecting the luminescent performance of the guests. We have added the above results to the revised manuscript.

Fig. R8. Fluorescence emission spectra of **BCZ** at different solvent (1.0×10^{-5} mol/L, Excitation wavelength: 340 nm).

Fig. R9. Fluorescence (a) and Phosphorescence (b) spectra of three doped materials

CBZ/PVP, CBZ/PVA, and CBZ/PA6.

Q10. In theoretical calculation, only data of **BCZ** and **BCZ-TPA** were presented. What about other molecules?

Response: Thanks to the reviewer for his/her suggestion. Due to the high computational workload of molecular dynamics and thermodynamics, we initially only selected **BCZ** with the weakest molecular motion ability and **BCZ-TPA** with the strongest molecular motion ability for theoretical calculations. Next, we conducted theoretical calculations on the other four guest molecules (**BCZ-Me**, **BCZ-nBu**, **BCZ-Be**, and **BCZ-Ph**) based on the suggestion of the reviewer. The theoretical calculation results shown that, from 293 K to 433 K, the torsional angle changes at positions 1 and 2 of four guest molecules also are very small (Fig. R10), and similar to **BCZ-TPA**, for the twist angles of **BCZ-Be** and **BCZ-Ph** at position 3, the magnitude of the torsion varies greatly with increasing temperature (Fig. R10). In addition, for the guests **BCZ-Me**, **BCZ-nBu**, **BCZ-Be**, and **BCZ-Ph**, the T*S (H-G) at 463 K was 73.4 J, 74.1 J, 86.4 J, and 101.5 J, respectively, and were 1.02 times, 1.03 times, 1.2 times, and 1.41 times that of **BCZ**, respectively (Fig. R11). We have added the calculation results in the revised manuscript.

Fig. R10. The distribution of the torsion angles of four guests (a, BCZ-Me; b, BCZ-*n*Bu; c, BCZ-Be; d, BCZ-Ph) at different temperatures.

Fig. R11. The thermodynamic parameters of guests (a, BCZ-Me; b, BCZ-*n*Bu; c, BCZ-Be; d, BCZ-Ph) at different temperatures.

Q11. The corresponding names of compounds in Supplementary Fig. 12 should be provided.

Response: Thanks very much to the reviewer for his/her careful review of the manuscript. We have provided the corresponding compound names.

Added: Fluorescence emission spectra of six guests (a, BCZ; b, BCZ-Me; c, BCZ-

*n*Bu; d, BCZ-Be; e, BCZ-Ph; f, BCZ-TPA) at different temperatures (DMSO as solvent, 1.0×10^{-5} mol/L, Excitation wavelength: 380 nm).

Q12. The doped materials were used to identify rescue workers and trapped personnel in fires, but the cyclicity of the HTP behavior was poor (Supplementary Fig. 9). Are there any related experiments on circulation in fire or smoke environment?

Response: Thanks to the reviewer for his/her suggestion. Doped materials have excellent thermal stability, and in fact, the process of preparing doped materials requires continuous heating at high temperatures (even without the protection of nitrogen) for 12 hours. In addition, the doped materials have been prepared for nearly 14 months, during which they have undergone various tests at high temperatures. We supplemented ten cyclic experiments of doped materials from high to low temperatures, and the results clearly showed that the phosphorescence emission of the doped materials showed almost no thermal loss (Fig. R12).

Fig. R12. Phosphorescence intensity (a) and lifetime (b) of BCZ/PVP doped material under temperature rise and fall cycles (Delayed time: 1 ms, Ex.: 380 nm).

We carefully checked the revised manuscript and used red words to note the revised sections. I am greatly appreciated to you for your kind help. Please contact me by email if you have any questions. I will make further revision if necessary.

Best Regards,

Xiang Ma, Ph. D. Professor

East China University of Science and Technology, Shanghai 200237, P. R. China

E-mail: maxiang@ecust.edu.cn

REVIEWER COMMENTS

Reviewer #1 (Remarks to the Author):

The authors have addressed my concerns. I recommend the publication of this work.

Reviewer #2 (Remarks to the Author):

The authors have conducted the related experiments and answered the questions from reviewers. It was claimed by the authors that no systematic strategy for preparing HTP materials was available. However, it has been reported by Xie et al. (Ref. 33) that the introduction of UPy moieties to construct hydrogen bonding networks could immobilize a certain degree of phosphors at high temperature, thus achieving HTP (Adv. Optical Mater. 2021, 9, 2100782). Additionally, their recent research (Ref. 32) also demonstrated that the covalent bond cross-linking networks and the hydrogen bonding networks formed by amide groups could restrict the motions and vibrations of phosphors, and make the polymer films exhibit ultrahigh temperature resistance for phosphorescence emission (Sci. China Chem. 2023, 66, 1161-1168). In Li's paper (Ref. 38), it was stated that the short-range π - π coupling contributed to the strong temperature resistance of phosphorescence emission (Angew. Chem. Int. Ed. 2023, 62, e202302792). So, several systematic strategies for preparing HTP materials have been proposed. There are still some mistakes in the revised manuscript. Some inconsistent data in the revised manuscript and response letter hinder the understanding of the results. The authors need to address and rectify any inconsistencies in their data to ensure the accuracy and reliability of their findings. And I have several questions about the added experimental details:

1. The authors stated that they had thoroughly checked and corrected the errors in the revised manuscript. But some evident errors are still there. For example, the space was omitted in "0.77s" (Line 152). In line 312, it should be "at 293 K" rather than "at 393 K" for the statement: "These two control materials also exhibit an ultra-long green afterglow with an emission wavelength of 485/517 nm for about 45 s at 293 K (Fig. 5a)". In Fig. 6a, the name of cBCZ should be "cBCZ" rather than "cBCZ-Me". In line 454, the reaction time was not provided.
2. In the revised Supplementary materials, some errors were also found. For example, in Supplementary Fig. 16 and Fig. 17, no Supplementary Fig. 16f and Fig. 17f could be found, although "f" was indicated in the captions. The words "CBZ/PVP, CBZ/PVA and CBZ/PA6" should be corrected in the caption of Supplementary Fig. 23. The temperature annotation in Supplementary Fig. 26 should be corrected.
3. The authors theoretically calculated the optimal configurations of six guest molecules in the gas phase state. But the calculated configuration of BCZ-Be in Fig. R3d and Supplementary Fig. 10d was different from the simulated configuration of it in Fig. 3a. The presence of inconsistent data can indeed lead to confusion and hinder the understanding of the results. It is important for the authors to address and rectify any inconsistencies in their data to ensure the accuracy and reliability of their findings.
4. BCZ and the control guests exhibited strong and persistent phosphorescence emission in solution at 77 K, indicating that when the strong motions of molecules were restricted, excellent phosphorescence

performance was obtained. However, in such a condition, the various phosphorescence lifetime and intensity of the molecules indicated that their electronic structures also had a significant influence on their phosphorescence performance, apart from the molecular motions. When the connected groups were changed from hydrogen atom to methyl and butyl groups, the guest molecules showed similar electronic structures with similar phosphorescence lifetime at 77 K. Though the motion ability of the molecules increased from BCZ to BCZ-nBu as revealed by theoretical calculations, minor differences in phosphorescence intensity decrease were found. It seems that the motion ability of the guest molecules does not have a great influence on the HTP performance. Moreover, the molecular orbital theory calculations revealed that BCZ-Ph and BCZ-TPA had different electronic structures from other molecules, which could also contribute to their poor HTP performance. Therefore, the selection of the control guests might not be reasonable enough to support the conclusion

5. Although the intensity of molecular motion directly affects its luminescence intensity, the introduction of fluorochromes and $n \rightarrow \pi^*$ transition systems also has a great influence on determining luminescence properties. Therefore, it is not appropriate to solely attribute the decrease in luminescence intensity from BCZ to BCZ-TPA in solution to the increased motion ability of the connected groups. The presence of different chromophores and changes in electronic structure can contribute to the observed decrease in luminescence intensity.

6. The determination of the T_g values based on the DSC curves of PVA and PA6 might not be reliable, as there was no clear, sudden downward shift observed. Additionally, it is important to provide the molecular weights of the polymers, as they can also influence the T_g values.

7. The normalized PL spectra, the phosphorescence Q.Y. and lifetimes of BCZ in different polymers illustrated that superior RTP performance was found in PVA and PA6 matrixes. These findings suggest that factors other than just the rigidity of the polymer chains could have a significant impact on the RTP and HTP performance.

8. Were cBCZ and cgBCZ commercially obtained or synthesized in the lab? The authors claimed that the compounds were designed and synthesized. Thus, the synthetic route should be provided. The HPLC and HRMS spectra of cBCZ, cgBCZ, cBCZ-Me and cgBCZ-Me should also be provided.

9. The phosphorescence Q.Y. and lifetimes of cBCZ/PVP, cBCZ-Me/PVP, cgBCZ/PVP and cgBCZ-Me/PVP at different temperatures should be measured.

10. In the response to Q10, the authors stated that for the guests BCZMe, BCZ-nBu, BCZ-Be, and BCZ-Ph, the T^*S (H-G) at 463 K was 73.4 J, 74.1 J, 86.4 J, and 101.5 J, respectively. It was different from the results presented in the revised manuscript (Line 290), which stated the values as 75 J, 83 J, 90 J, and 98 J, respectively. These inconsistent data also led to confusion and hindered the understanding of the results. The T^*S (H-G) value of BCZ-nBu at 463 K was 1.15 times higher than that of BCZ. However, minor differences in HTP performance were found for the two compounds, which also indicated that molecular motion was not the only factor that determined HTP performance.

Manuscript ID: NCOMMS-23-33760A

TITLE: Twofold Rigidity Activates Ultralong Organic High-Temperature Phosphorescence

Dear Dr. A. Patterson,

We have received your letter about our manuscript and carefully read the reviewer's comments in the e-mail. We thank you again for giving us the opportunity to consider our manuscript. We have carefully revised our manuscript according to the suggestions of you and the reviewer, and the red font in the revised manuscript is the revised part. The point-to-point answers to the reviewer's questions are as follows.

Reviewer #1 (Remarks to the Author):

The authors have addressed my concerns. I recommend the publication of this work.

Response: Thanks to the reviewer for his/her support.

Reviewer #2 (Remarks to the Author):

The authors have conducted the related experiments and answered the questions from reviewers. It was claimed by the authors that no systematic strategy for preparing HTP materials was available. However, it has been reported by Xie et al. (Ref. 33) that the introduction of UPy moieties to construct hydrogen bonding networks could immobilize a certain degree of phosphors at high temperature, thus achieving HTP (Adv. Optical Mater. 2021, 9, 2100782). Additionally, their recent research (Ref. 32) also demonstrated that the covalent bond cross-linking networks and the hydrogen bonding networks formed by amide groups could restrict the motions and vibrations of phosphors, and make the polymer films exhibit ultrahigh temperature resistance for phosphorescence emission (Sci. China Chem. 2023, 66, 1161-1168). In Li's paper (Ref. 38), it was stated that the short-range π - π coupling contributed to the strong temperature resistance of phosphorescence emission (Angew. Chem. Int. Ed. 2023, 62, e202302792). So, several systematic strategies for preparing HTP materials have been proposed. There are still some mistakes in the revised manuscript. Some inconsistent data in the revised manuscript and response letter hinder the understanding of the results. The authors need to address and rectify any inconsistencies in their data to ensure the accuracy and reliability of their findings. And I have several questions about the added experimental details:

Response: We are very grateful to the reviewer for his/her careful review of the manuscript, and we sincerely apologize for our negligence. The seriousness and professionalism of this reviewer have greatly helped to improve the quality of the

manuscript. We will avoid these errors in our future work and once again express our respect to the reviewer. We have made revision to the statement in the article regarding “no systematic strategy for preparing HTP materials” based on the reviewer's suggestions.

Added in the revised manuscript: “The aforementioned findings provide important references for designing HTP materials. However, it remains a challenge to propose facile and feasible molecular design strategy for preparing HTP materials to further extend the application potential.”

Q1. The authors stated that they had thoroughly checked and corrected the errors in the revised manuscript. But some evident errors are still there. For example, the space was omitted in “0.77s” (Line 152). In line 312, it should be “at 293 K” rather than “at 393 K” for the statement: “These two control materials also exhibit an ultra-long green afterglow with an emission wavelength of 485/517 nm for about 45 s at 293 K (Fig. 5a)”. In Fig. 6a, the name of **cBCZ** should be “**cBCZ**” rather than “**cBCZ-Me**”. In line 454, the reaction time was not provided.

Response: Thanks very much for the careful review by the reviewer. We have checked and corrected the errors in the revised manuscript.

Q2. In the revised Supplementary materials, some errors were also found. For example, in Supplementary Fig. 16 and Fig. 17, no Supplementary Fig. 16f and Fig. 17f could be found, although “f” was indicated in the captions. The words “**CBZ/PVP**, **CBZ/PVA** and **CBZ/PA66**” should be corrected in the caption of Supplementary Fig. 23. The temperature annotation in Supplementary Fig. 26 should be corrected.

Response: We are sorry for the mistakes. We have corrected them in the revised manuscript.

Q3. The authors theoretically calculated the optimal configurations of six guest molecules in the gas phase state. But the calculated configuration of **BCZ-Be** in Fig. R3d and Supplementary Fig. 10d was different from the simulated configuration of it in Fig. 3a. The presence of inconsistent data can indeed lead to confusion and hinder the understanding of the results. It is important for the authors to address and rectify any inconsistencies in their data to ensure the accuracy and reliability of their findings.

Response: Thanks to the reviewer for his/her careful review. We sincerely apologize for the mistake. We have carefully checked the data and found that the results reflected in the supporting information are correct. We have replaced the data in Fig. 3a.

Q4. **BCZ** and the control guests exhibited strong and persistent phosphorescence emission in solution at 77 K, indicating that when the strong motions of molecules were restricted, excellent phosphorescence performance was obtained. However, in such a condition, the various phosphorescence lifetime and intensity of the molecules indicated that their electronic structures also had a significant influence on their phosphorescence performance, apart from the molecular motions. When the connected groups were changed from hydrogen atom to methyl and butyl groups, the guest molecules showed similar electronic structures with similar phosphorescence lifetime at 77 K. Though the motion ability of the molecules increased from **BCZ** to **BCZ-*n*Bu** as revealed by theoretical calculations, minor differences in phosphorescence intensity decrease were found. It seems that the motion ability of the guest molecules does not have a great influence on the HTP performance. Moreover, the molecular orbital theory calculations revealed that **BCZ-Ph** and **BCZ-TPA** had different electronic structures from other molecules, which could also contribute to their poor HTP performance. Therefore, the selection of the control guests might not be reasonable enough to support the conclusion

Response: Thanks to the reviewer for his/her suggestions.

Please allow us to briefly summarize the entire text. In this work, six guest molecules were designed and synthesized, among which **BCZ**, **BCZ-Me**, and **BCZ-*n*Bu** have similar and best luminescent properties, while the luminescent performance of **BCZ-Be**, **BCZ-Ph**, and **BCZ-TPA** gradually decreases. Such as, the fluorescence emission intensity, high temperature resistance of fluorescence, and low-temperature phosphorescence lifetime of six guests all conform to this trend (Due to the significant restriction of the movement of molecules at low temperature, the weaker motion ability of six guests results in their phosphorescence lifetime only decreasing from 6.3 s to 4.8 s). Next, six corresponding doped materials were prepared, and their luminescent properties were similar to those of the guests, **BCZ/PVP**, **BCZ-Me/PVP**, and **BCZ-*n*Bu/PVP** have similar and best HTP performance, and the HTP performance of **BCZ-Be/PVP**, **BCZ-Ph/PVP**, and **BCZ-TPA/PVP** gradually decreases.

As stated by the reviewer, three guests **BCZ**, **BCZ-Me**, and **BCZ-*n*Bu** exhibit similar luminescent properties. Especially **BCZ-Me** and **BCZ-*n*Bu** have almost identical absorption and emission spectra (Fig. R1). This may be because when the number of carbon atoms in a linear alkyl group is greater than two, subsequent carbon atoms are almost unable to form effective conjugation with the molecule, and it is also difficult to affect the electronic structure of the molecule, thus having little effect on the excited state property of the molecule, so the influence of the motion of alkyl chain groups on the radiative transition of excitons can be almost negligible. Therefore, for three guests (**BCZ**, **BCZ-Me**, and **BCZ-*n*Bu**), the decrease in fluorescence intensity

is minimal as the temperature increases (Fig. 3c, Fig. R2). In addition, the theoretical calculations shown that, at 293 K, the $T \times S$ values of **BCZ-Me** (31 J) and **BCZ-*n*Bu** (32 J) were only 1.03 times and 1.06 times higher than that of **BCZ** (30 J), respectively; and at 463 K, the values of **BCZ-Me** (75 J) and **BCZ-*n*Bu** (83 J) were 1.04 times and 1.15 times higher than that of **BCZ** (72 J), respectively. (For the value of **BCZ-*n*Bu** was 1.15 times higher than that of **BCZ**, this is due to the presence of *n*-butyl group. The alkyl chains have excellent flexibility, so they are more prone to disorderly motion at higher temperatures, which increases the entropy change of molecules, resulting in increase in $T \times S$ values. However, as described above, this motion hardly affects the excited state luminescent property of the molecules, so the three guests **BCZ**, **BCZ-Me**, and **BCZ-*n*Bu** have similar luminescent properties.)

For other three guests **BCZ-Be**, **BCZ-Ph**, and **BCZ-TPA**, the benzyl, benzene, or triphenylamine groups can form conjugation with the dibenzo[*a, c*]carbazole moiety of the molecules. From guests **BCZ** to **BCZ-TPA**, the fluorescence emission wavelength and phosphorescence emission wavelength of the molecules did undergo a red shift (Q5, Fig. R3), but the degree of change was relatively small. In addition, although various groups were connected to the **CBZ** molecule to obtain the other five guests, the overall shape of the spectra of six guests is very similar, which indicating that the luminescent core of the six guests is still the dibenzo[*a, c*]carbazole moiety. However, it is precisely because the three groups can form conjugation with molecules that they can act as rotors to consume excited state energy, resulting in a decrease in the phosphorescence performance of doped materials (*Chem. Eur. J.* 2018, 24, 14269).

Fig. R1. (a) UV absorption spectra, (b) fluorescence emission spectra (THF as solvent, 1.0×10^{-5} mol/L, Ex.: 340 nm), and (c) phosphorescence emission spectra of guests **BCZ-Me** and **BCZ-*n*Bu** (THF as solvent, 1.0×10^{-5} mol/L, 77 K, Ex.: 380 nm, Delayed time: 1ms).

Fig. R2. Relative changes in fluorescence intensity of six guests at various temperatures (*p*-xylene as solvent, 1.0×10^{-5} mol/L).

Q5. Although the intensity of molecular motion directly affects its luminescence intensity, the introduction of fluorochromes and $n \rightarrow \pi^*$ transition systems also has a great influence on determining luminescence properties. Therefore, it is not appropriate to solely attribute the decrease in luminescence intensity from **BCZ** to **BCZ-TPA** in solution to the increased motion ability of the connected groups. The presence of different chromophores and changes in electronic structure can contribute to the observed decrease in luminescence intensity.

Response: Thanks to the reviewer for his/her suggestions. As the reviewer pointed out, the introduction of fluorochromes and $n \rightarrow \pi^*$ transition systems may alter the luminescent properties of molecules, such as excited state energy levels and emission wavelengths. As shown in Fig. R3, from guests **BCZ** to **BCZ-TPA**, the fluorescence emission wavelength and phosphorescence emission wavelength of the molecules did undergo a red shift, but the degree of change was relatively small, additionally, the overall shape of the spectra is also very similar. These results indicate that the introduction of three groups (benzyl, benzene, and triphenylamine groups) has a relatively small impact on the molecular excited-state properties. This may be because the core luminescent unit of the six guests is the planar dibenzo[*a, c*]carbazole moiety, and the main function of the benzyl, benzene, or triphenylamine groups is to act as a rotor to consume excited state energy. Therefore, we believe that the thermal motion of molecules may be the main reason for reducing the HTP performance of doped materials. However, we still greatly appreciate the different insights provided by the reviewer, which have expanded our research approach in the upcoming work.

Fig. R3. (a) Fluorescence emission spectra of six guests (THF as solvent, 1.0×10^{-5} mol/L, Ex.: 340 nm). (b) Phosphorescence emission spectra of six guests (THF as solvent, 1.0×10^{-5} mol/L, 77 K, Ex.: 380 nm, Delayed time: 1ms).

Q6. The determination of the T_g values based on the DSC curves of might not be reliable, as there was no clear, sudden downward shift observed. Additionally, it is important to provide the molecular weights of the polymers, as they can also influence the T_g values.

Response: Thanks to the reviewer for his/her suggestions. We have provided the molecular weights of the polymers in the revised manuscript. To ensure the accuracy of the T_g values of two polymers (**PVA** and **PA66**), we chose a more precise instrument and further optimized the testing conditions. The results shown that the obtained T_g values were consistent with the previous results (Fig. R4). In addition, the performing autofitting T_g values also match the reported by other researchers (*Nat. Commun.* 2021, 12, 2297; *Angew. Chem. Int. Ed.* 2023, e202300927; *Polymers*, 2022, 14, 442; *Polymers*, 2022, 14, 3879).

Added in the revised manuscript: **PVP** (molecular weight: 360000 g/mol), **PVA** (molecular weight: 85000 to 124000 g/mol), and **PA66** (molecular weight: 43000 g/mol).

Fig. R4. DSC curves of (a) PVA and (b) PA66.

Q7. The normalized PL spectra, the phosphorescence Q.Y. and lifetimes of **BCZ** in different polymers illustrated that superior RTP performance was found in **PVA** and **PA66** matrixes. These findings suggest that factors other than just the rigidity of the polymer chains could have a significant impact on the RTP and HTP performance.

Response: Thanks to the reviewer for his/her suggestions. At 293 K, the phosphorescence lifetimes of the three doped materials (**BCZ/PVP**, **BCZ/PVA**, and **BCZ/PA66**) were 4.18 s, 5.09 s, and 4.94 s, respectively; and the phosphorescence Q.Y. values were 22.4%, 27.6%, and 24.3%, respectively. Doped material **BCZ/PVA** has relatively optimal RTP performance, which may be due to the strong curing ability of **PVA** matrix. However, from 293 K to 373 K, the phosphorescence lifetimes of three doped materials decreased by 48%, 75%, and 86%, respectively (Fig. 5g); the phosphorescence Q.Y. values decreased by 59%, 81%, and 93%, respectively (Fig. 5h), and as the temperature continued to rise, the afterglow of **BCZ/PA66** first disappeared, followed by that of **BCZ/PVA**, and finally the afterglow of **BCZ/PVP** disappeared. Among three doped materials, although the RTP performance of **BCZ/PVP** is relatively weak, it has a significantly optimal HTP performance, followed by HTP performance of **BCZ/PVA**, and the HTP performance of **BCZ/PA66** is the weakest. This may be because the host **PVP** has the highest T_g value (456 K), followed by **PVA** (355 K), and finally **PA66** (326 K). When the temperature is higher than the T_g values of the polymers, the rigid curing ability of the polymers will decrease, indicating that the host have lost the ability to inhibit the motion of guest molecules. Therefore, the phosphorescence performance of doped materials will decrease until it disappears.

Q8. Were **cBCZ** and **cgBCZ** commercially obtained or synthesized in the lab? The authors claimed that the compounds were designed and synthesized. Thus, the synthetic route should be provided. The HPLC and HRMS spectra of **cBCZ**, **cgBCZ**, **cBCZ–Me** and **cgBCZ–Me** should also be provided.

Response: Thanks to the reviewer for his/her suggestions. Guests **cBCZ** and **cgBCZ** were commercially purchased, but they need to be purified through column chromatography before use. We have tested the high-performance liquid chromatography and high-resolution mass spectrometry of four guests (**cBCZ**, **cgBCZ**, **cBCZ–Me** and **cgBCZ–Me**) (Fig. R5, Fig. R6).

Added in the revised manuscript: *7H*-benzo[*c*]carbazole (**cBCZ**), *7H*-dibenzo[*c,g*]carbazole (**cgBCZ**), 7-methyl-*7H*-benzo[*c*]carbazole (**cBCZ–Me**), and 7-methyl-*7H*-dibenzo[*c,g*]carbazole (**cgBCZ–Me**), were selected or synthesized as guests to construct doped systems with **PVP** as the host (**cBCZ** and **cgBCZ** were commercially purchased but require further purification through column chromatography) (Fig. 6a). The molecular structures and purities of four guests were

confirmed by single-crystal X-ray diffraction, high-resolution mass spectroscopy, nuclear magnetic resonance spectroscopy, and high-performance liquid chromatography (Supplementary Fig. 26).

Fig. R5. HPLC spectrometry of four guests (a) **cBCZ**, (b) **cBCZ-Me**, (c) **cgBCZ**, and (d) **cgBCZ-Me** ($\text{CH}_3\text{OH}/\text{hexane} = 70\%: 30\%$).

Fig. R6. High-resolution mass spectrometry of four guests (a) **cBCZ**, (b) **cBCZ-Me**, (c) **cgBCZ**, and (d) **cgBCZ-Me**.

Q9. The phosphorescence Q.Y. and lifetimes of **cBCZ/PVP**, **cBCZ-Me/PVP**, **cgBCZ/PVP** and **cgBCZ-Me/PVP** at different temperatures should be measured.

Response: Thanks to the reviewer for his/her suggestions. We have measured the phosphorescence Q.Y. and phosphorescence lifetime of four doped materials at different temperatures (Fig. R7).

Added in the revised manuscript: The phosphorescence lifetime of **cBCZ/PVP** were 2.32 s, 0.91 s, and 0.40 s at 293 K, 373 K, and 413 K, respectively (Supplementary Fig. 28a), and those of **cBCZ-Me/PVP** were 2.28 s, 0.88 s, and 0.38 s, respectively (Supplementary Fig. 28b). The phosphorescence Q.Y. values of **cBCZ/PVP** were 16.4%, 8.2%, and 2.5% at 293 K, 373 K, and 413 K, respectively, and those of **cBCZ-Me/PVP** were 17.0%, 7.8%, and 2.3%, respectively. Whereas the corresponding afterglow times of **cgBCZ/PVP** and **cgBCZ-Me/PVP** were approximately 11 s, 6 s, and 2 s, respectively (Fig. 6d). The phosphorescence lifetime of **cgBCZ/PVP** were 1.36 s, 0.63 s, and 0.21 s at 293 K, 373 K, and 413 K, respectively (Supplementary Fig. 28c), and those of **cgBCZ-Me/PVP** were 1.40 s, 0.66 s, and 0.23 s, respectively (Supplementary Fig. 28d). The phosphorescence Q.Y. values of **cgBCZ/PVP** were 14.1%, 6.8%, and 2.0% at 293 K, 373 K, and 413 K, respectively, and those of **cgBCZ-Me/PVP** were 13.8%, 6.4%, and 2.1%, respectively.

Fig. R7. Kinetic attenuation curves of four doped materials (a) **cBCZ/PVP**, (b) **cgBCZ/PVP**, (c) **cBCZ-Me/PVP**, and (d) **cgBCZ-Me/PVP** at different temperatures (Ex.: 380 nm).

Q10. In the response to Q10, the authors stated that for the guests **BCZ-Me**, **BCZ-*n*Bu**, **BCZ-Be**, and **BCZ-Ph**, the $T \times S$ (H-G) at 463 K was 73.4 J, 74.1 J, 86.4 J, and 101.5 J, respectively. It was different from the results presented in the revised manuscript (Line 290), which stated the values as 75 J, 83 J, 90 J, and 98 J, respectively. These inconsistent data also led to confusion and hindered the understanding of the results. The $T \times S$ (H-G) value of **BCZ-*n*Bu** at 463 K was 1.15 times higher than that of **BCZ**. However, minor differences in HTP performance were found for the two compounds, which also indicated that molecular motion was not the only factor that determined HTP performance.

Response: Thanks to the reviewer for his/her careful review. We sincerely apologize for the mistake. As shown in Supplementary Fig. 21, for the guests **BCZ-Me**, **BCZ-*n*Bu**, **BCZ-Be**, and **BCZ-Ph**, the $T \times S$ values at 463 K were 75 J, 83 J, 90 J, and 98 J, respectively.

For the $T \times S$ (H-G) value of **BCZ-*n*Bu** at 463 K was 1.15 times higher than that of **BCZ**, we think that this is due to the presence of *n*-butyl group. For the alkyl chains, due to their flexibility, they are more prone to disorderly motion at higher temperatures, which increases the entropy change of molecules, and resulting in increase in $T \times S$ values. However, this motion hardly affects the excited state luminescent property of the molecules. Indeed, at 293 K, the $T \times S$ value of **BCZ-*n*Bu** (32 J) was 1.06 times higher than that of **BCZ** (30 J). In addition, the $T \times S$ values of **BCZ-Me** at 293 K and 463 K were 1.03 times and 1.04 times higher than that of **BCZ**, respectively.

Finally, after two rounds of review by the reviewers, the accuracy and completeness of the paper have been greatly improved. We once again express our sincere gratitude to the reviewers. Although there may be shortcomings in the paper, we have made effort to solve scientific problems related to organic high-temperature phosphorescence materials. We will also make further efforts to research and develop organic HTP materials in future work.

Best Regards,

Xiang Ma, Ph. D. Professor
School of Chemistry and Molecular Engineering,
East China University of Science and Technology, Shanghai 200237, P. R. China
E-mail: maxiang@ecust.edu.cn

REVIEWERS' COMMENTS

Reviewer #2 (Remarks to the Author):

I am pleased to see that the authors have incorporated the suggested revisions and have provided additional information where necessary, enhancing the clarity of the work. The manuscript is recommended for publication.

Manuscript ID: NCOMMS-23-33760B

TITLE: Twofold Rigidity Activates Ultralong Organic High-Temperature Phosphorescence

Reviewer #2 (Remarks to the Author):

I am pleased to see that the authors have incorporated the suggested revisions and have provided additional information where necessary, enhancing the clarity of the work. The manuscript is recommended for publication.

Response: Thanks to the reviewer for his/her support.